# DyCAST: Learning Dynamic Causal Structure from Time Series

**Yue Cheng**[1]*    **Bochen Lyu**[2,3]    **Weiwei Xing**[1,✉]    **Zhanxing Zhu**[3,✉]
[1]Beijing Jiaotong University    [2]DataCanvas    [3]University of Southampton
yuecheng@bjtu.edu.cn   bochen.lv@gmail.com
wwxing@bjtu.edu.cn   z.zhu@soton.ac.uk

## ABSTRACT

Understanding the dynamics of causal structures is crucial for uncovering the underlying processes in time series data. Previous approaches rely on static assumptions, where contemporaneous and time-lagged dependencies are assumed to have invariant topological structures. However, these models fail to capture the evolving causal relationship between variables when the underlying process exhibits such dynamics. To address this limitation, we propose DyCAST, a novel framework designed to learn dynamic causal structures in time series using Neural Ordinary Differential Equations (Neural ODEs). The key innovation lies in modeling the temporal dynamics of the contemporaneous structure, drawing inspiration from recent advances in Neural ODEs on constrained manifolds. We reformulate the task of learning causal structures at each time step as solving the solution trajectory of a Neural ODE on the directed acyclic graph (DAG) manifold. To accommodate high-dimensional causal structures, we extend DyCAST by learning the temporal dynamics of the hidden state for contemporaneous causal structure. Experiments on both synthetic and real-world datasets demonstrate that DyCAST achieves superior or comparable performance compared to existing causal discovery models.

## 1 INTRODUCTION

Learning causal structures from time series data has been recognized as a fundamental and challenging problem due to its widespread use in various domains, such as traffic (Cheng et al., 2024b), biology (Sachs et al., 2005; Yu et al., 2023), healthcare (Lucas et al., 2004) etc. Recently, Pamfil et al. (2020); Sun et al. (2023); Gao et al. (2022) have made significant efforts in this direction of causal discovery through the *directed acyclic graphs* (DAGs), which provide important clues about the relationship between variables particularly in a multivariate dynamical system.

Existing methods typically adhere DYNOTEARS (Pamfil et al., 2020) to a paradigm that classifies causal structures in time series into contemporaneous (*intra-slice*) and time-lagged (*inter-slice*) dependencies. As shown in Figure. 1 (b), the weight of each edge between nodes (drawn as solid lines) at the same time $t$ represents the intra-slice dependencies, while the weight of edges between nodes (drawn as dashed lines) at different times captures inter-slice. However, most such results (Pamfil et al., 2020; Sun et al., 2023; Li et al., 2024) mainly focused on *static DAGs*, i.e. intra- and inter-slice structures with invariant topology over a given set of variables. A major challenge in dynamic data in real-world is its inherently non-stationary properties, in which DAG topological structures (i.e. variables' causality) can evolve over time. Examples include traffic networks, where roads may have periodically variable lanes, and the network can undergo significant changes due to sudden accidents or temporary traffic controls (For more details, please refer to the visualization results in Section 4.3). In these scenarios, modeling evolutionary intra-slice causal patterns is crucial for accurately capturing the underlying dynamic properties of the system.

A significant challenge in learning dynamic causal structures is that the edges in DAGs can emerge or disappear over time while still maintaining the directed acyclic property, see Figure. 1 (c) for the

---

*Contributed during an internship at DataCanvas.

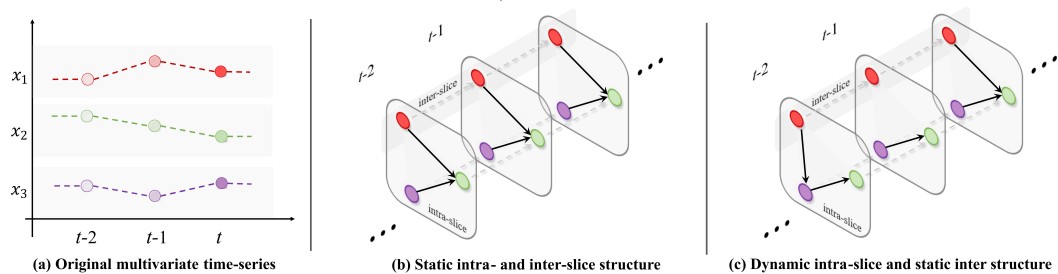

Figure 1: Illustration of the intra-slice (solid lines) and inter-slice (dashed lines) dependencies in a SEM with $d = 3$ variables and autoregression order $p = 1$ under different dynamical settings. (a) The original multivariate time series. (b) The causal discovery framework with a static setting. (c) Our causal discovery framework with dynamic intra-slice dependencies and static inter-slice dependencies settings.

dynamic of the DAGs across time. However, current methodologies are still insufficient for modeling such a dynamic causal structure. More recent efforts have concentrated on recovering *dynamic inter-slice structures*, as their continuous evolution avoids the hard constraints of maintaining a DAG structure (Song et al., 2009). For instance, Gao & Yang (2022) proposed a time-varying Granger causal network, where the inter-slice structures are learned via kernel-reweighted group lasso; such approach requires prior knowledge for appropriately selecting the kernel and does not model intra-slice dependencies. Nevertheless, we still expect to recover *dynamic intra-slice structures* from multivariate time series. This is the main goal of our work.

To this end, we propose **DyCAST**, a **dy**namic **ca**usal **st**ructure learning approach for time series based on Neural Ordinary Differential Equations (Neural ODEs). Targeting the dynamic intra-slice structures, we devise a constrained Neural ODE to model the temporal latent states dynamics of DAGs. See Figure. 1 for the difference between previous work and our proposed DyCAST. Compared with DYNOTEARS, we enforce DAG manifold constraint for the dynamics function to model how the change of intra-slice structure over time. Further, the dynamic function is transformed into latent space to model the complex structure's change for high-dimension time series. Empirical results across synthetic dynamic and real world datasets substantiate the effectiveness of DyCAST.

## RELATED WORK

**Static causal structures learning.** Inspired by the success of NOTEARS-based methods of causal discovery (Zheng et al., 2018), a series of NOTEARS variants were proposed to recover the causal structures in time series. Specifically, DYNOTEARS (Pamfil et al., 2020) pioneered extended instantaneous linear NOTEARS exploiting autoregression. NTS-NOTEARS (Sun et al., 2023) proposed to capture nonlinear relations with 1-D CNN. IDYNO (Gao et al., 2022) and TECDI (Li et al., 2023) incorporated intervention data into the DYNOTEARS framework to enhance identifiability. However, the above existing methods for learning temporal causal structure typically assume that the intra-slice and inter-slice topologies remain static.

**Dynamic causal structures learning.** With the assumption that the causal structures are varing across time, works used various methods for recovering dynamic structures. TVDBN (Song et al., 2009) and KWgL (Gao & Yang, 2022) proposed to adopt kernel-reweighted $\ell_1$-regularized autoregressive procedure for modeling the structurally varying. TVGL (Hallac et al., 2017) derived a scalable message-passing algorithm to infer time-varying structures. However, these works only model the dynamics of inter-slice, and none of them address the dynamics of intra-slice.

**Neural ODEs.** Neural ODEs were originally introduced to model time series with continuous latent dynamics, offering a natural way to capture the dynamical behavior of systems (Chen et al., 2018). However, they have not been extensively explored in the context of dynamic causal structures, as most prior work has applied them to model node interactions in time series forecasting tasks. For instance, HOPE (Luo et al., 2023) incorporated second-order graph ODEs to model higher-order temporal dependencies, while GDERec implicitly captures the temporal evolution of user-item in-

teraction graphs using GNN-based ODEs. SGODE (Chen et al., 2024) extended signed graph Neural ODEs to leverage both positive and negative node information during continuous dynamics. LCCM (Brouwer et al., 2021) leveraged latent dynamics for causal discovery but is limited to static or quasi-static relationships, overlooking dynamic intra-slice structures. Despite these advancements, most existing models focus primarily on time series forecasting, rather than addressing changes in intra-slice causal structures.

## 2 PRELIMINARIES

**Notations.** We let $\mathbb{D}$ denote the space of DAGs on $d$ variables. Given $N$ independent realizations of a multivariate time series represented as $\mathcal{X} = \{\boldsymbol{X}_{:,0}, \boldsymbol{X}_{:,1}, \cdots, \boldsymbol{X}_{:,t}, \cdots, \boldsymbol{X}_{:,T}\}$, where $\boldsymbol{X}_{:,t} = \{\boldsymbol{x}^1_{1:N,t}, \boldsymbol{x}^2_{1:N,t}, \cdots, \boldsymbol{x}^i_{1:N,t}, \cdots, \boldsymbol{x}^d_{1:N,t}\} \in \mathbb{R}^{N \times d}$ is the recording of $d$ variables at time step $t$, where $d$ represents the number of variables in the dataset. With a slight abuse of notation, we will use $\boldsymbol{X}_t$ interchangeably.

**Causal structure model for time series.** Let $\boldsymbol{Y} = [\boldsymbol{X}_{t-1}|\cdots|\boldsymbol{X}_{t-p}]$ be the $N \times pd$ matrix of $p$-order time-lagged version of $\boldsymbol{X}_t$, $\boldsymbol{W}$ be the $d \times d$ matrix of intra-slice and $\boldsymbol{A} = [\boldsymbol{A}_1^T|\cdots|\boldsymbol{A}_p^T]^T$ be the $pd \times d$ matrix of inter-slice. A causal structure model for time series can be equivalently represented by a structural equation model (SEM) (Pamfil et al., 2020):

$$\boldsymbol{X}_t = \boldsymbol{X}_t \boldsymbol{W} + \boldsymbol{Y} \boldsymbol{A} + \boldsymbol{Z}_t, \tag{1}$$

where $t \in \{1, 2, \cdots, T\}$ with horizon $T$, $\boldsymbol{Z}_t \sim \mathcal{N}(0, \sigma^2 I)$ is a noise vector with independent elements across time. The intra-slice structure is defined by the nonzero elements in $\boldsymbol{W}$, i.e. $\boldsymbol{X}_t^i$ is dependent on $\boldsymbol{X}_t^j$ if and only if the coefficient $\boldsymbol{W}^{ij}$ is nonzero. The inter-slice structure is defined by the nonzero elements in $\boldsymbol{A}_k$, i.e. $\boldsymbol{X}_t^i$ is dependent on $\boldsymbol{X}_{t-k}^j$ if and only if the $\boldsymbol{A}_k^{ij}$ is nonzero.

**Neural ODEs.** Neural ODEs (Chen et al., 2018) are a class of deep learning models that effectively learn temporal dynamics for time series modeling. They use a neural network to map $x_0$ into a hidden space $z_0$, where a continuous model $f_\theta$ governs the dynamics, described with following:

$$z_t = z_0 + \int_0^t f_\theta(z_s)\, ds \text{ where } z_0 = \zeta(x_0), \text{ and } x_t = \xi(z_t), \tag{2}$$

where $\xi$ and $\zeta$ are parametrized neural networks that model the relationship $x_0 \mapsto z_0$ and $\boldsymbol{z}_t \mapsto x_t$, respectively. The dynamics of the hidden state, $f_\theta$, are specified by another learnable neural network.

## 3 LEARNING DYNAMIC CAUSAL STRUCTURE

### 3.1 PROBLEM DEFINITION

We will target recovering the dynamic causal structures. That is, the intra-slice dependencies between variables in time $t$ and $t-1$ is not fixed, and $\boldsymbol{W}$ changes over time. Following the practice in causal structure discovery (Pamfil et al., 2020), we formulate the problem as finding a series snapshots of DAG $\mathcal{W} = \{\mathcal{W}_0, \cdots, \mathcal{W}_T\}$ to capture the time-dependent causal structure, and a series of graph $\mathcal{A} = \{\mathcal{A}_1, \cdots, \mathcal{A}_k, \cdots, \mathcal{A}_p\}$ for time-lagged. Each graph $\mathcal{W}_t = (\mathcal{V}, \mathcal{E}_t, \boldsymbol{W}_t)$ or $\mathcal{A}_k = (\mathcal{V}, \mathcal{E}, \boldsymbol{A}_k)$ is a weighted graph with a shared variable set $\mathcal{V}$, a causal link set $\mathcal{E}$, and weighted adjacency matrix $\boldsymbol{W}_t \in \mathbb{R}^{d \times d}$ or $\boldsymbol{A}_k \in \mathbb{R}^{d \times d}$. Thus, a linear form of the dynamic causal discovery model in SEM is:

$$\boldsymbol{X}_t = \boldsymbol{X}_t \boldsymbol{W}_t + \boldsymbol{Y} \boldsymbol{A} + \boldsymbol{Z}_t. \tag{3}$$

In order to accurately manipulate the time-dependent causal structure of time series, we use a smooth dynamics assumption to characterize the changes in intra-slice structure. Then, we further find a function $F_\theta$ to model the evolution of dynamic causal structure,

$$\boldsymbol{W}_t = F_\theta(t, \boldsymbol{W}_0) \ s.t. \ \boldsymbol{W}_t \in \mathbb{D}. \tag{4}$$

Thus, minimizing the least-squares loss under the DAG constraint leads to the following optimization problem:

$$\underset{\theta, A}{\arg\min} \mathcal{L}(\boldsymbol{W}_t(\theta), \boldsymbol{A}) = \frac{1}{2NT} \sum_{t=0}^{T} \|\boldsymbol{X}_t - \boldsymbol{X}_t \boldsymbol{W}_t(\theta) - \boldsymbol{Y}\boldsymbol{A}\|_2^2 \ s.t. \ \boldsymbol{W}_t \in \mathbb{D}. \tag{5}$$

## 3.2 CONSTRAINED NEURAL ODE FOR DYNAMIC DAGS

The change of intra-slice can reflect the underlying dynamics of time series data. To fully capture the dynamic causal structure, inspired by the idea of Neural ODEs, we propose to adopt a continuous dynamical system $f_\theta$ for parameterizing the function $F_\theta$, i.e.

$$\boldsymbol{W}_t = \boldsymbol{W}_0 + \int_0^t f_\theta(\boldsymbol{W}_s, s)\, ds,\ s \in [0, t]\ \ s.t.\ \ \boldsymbol{W}_s \in \mathbb{D}. \tag{6}$$

where $\boldsymbol{W}_0$ is also a learnable matrix that denotes the intra-slice matrix when time $t = 0$; $f_\theta(\cdot)$ specifies the dynamics of time-dependent intra-slice structure and is a neural network to be learned.

Since our setting introduces additional complexity compared to DYNOTEARS, we found that the difficulty in solving the dynamical system in Eq. (6) is the directed acyclicity constraint on $\boldsymbol{W}_s$. We are confronted with a primary question: *how to guarantee the solution trajectory $\boldsymbol{W}_s$ for the Neural ODE governed by $f_\theta$ consistently satisfies the constraint?*

**Dynamic acyclicity constraint.** We adopt an equivalent formulation via the trace exponential function studied in (Zheng et al., 2018) to characterize the DAG constraint,

$$h(\boldsymbol{W}) = \mathrm{tr}(e^{\boldsymbol{W} \circ \boldsymbol{W}}) - d \tag{7}$$

where "∘" denotes the Hadamard product of two matrices. They show that the function $h$ satisfies $h(\boldsymbol{W}) = 0$ if and only if $\boldsymbol{W}$ is acyclic. Replacing the acyclicity constraint with the equality constraint $h(\boldsymbol{W}) = 0$, we can regard the Neural ODE Eq. (6) as an underlying ODE (Rheinboldt, 1984) to address the main concern. We further reformulate the vector field $f_\theta$ with an invariant manifold $\mathcal{M} = \{\boldsymbol{W}_s \in \mathbb{R}^{d \times d}, h(\boldsymbol{W}) = 0\}$ as a constrainted Neural ODE (i.e. underlying Neural ODE),

$$\boldsymbol{W}_t = \boldsymbol{W}_0 + \int_0^t f_\theta(\boldsymbol{W}_s, s) - \gamma S(\boldsymbol{W}_s) h(\boldsymbol{W}_s)\, ds,\ s \in [0, t] \tag{8}$$

where $\gamma \geq 0$ is a scalar parameter, $S(\cdot) : \mathbb{R}^{d \times d} \to \mathbb{R}^{d \times d}$ in our paper is a stabilization matrix to guarantee the manifold $\mathcal{M}$ constraint gradually and consistently satisfied. Note that the underlying Neural ODE Eq.(8) is equivalent to the neural ODE Eq.(6) on the manifold $\mathcal{M}$ in the sense that they have the same (analytical) solution set for $\boldsymbol{W}_s$ on $\mathcal{M}$. By controlling the trajectory of the dynamic causal structure to asymptotically satisfy DAG constraints, hard enforcement techniques, such as the augmented Lagrangian, are unnecessary.

Inspired by White et al. (2023), we choose the Moore-Penrose pseudoinverse of the Jacobian matrix $G(\boldsymbol{W})$ of function $h$ as the stabilization matrix, where $G(\boldsymbol{W}) = \nabla h(\boldsymbol{W}) = (e^{\boldsymbol{W} \circ \boldsymbol{W}})^T \circ 2\boldsymbol{W}$. This matrix aims to guarantee that $S(\boldsymbol{W})G(\boldsymbol{W})$ is symmetric positive definite with the smallest eigenvalue bounded away from zero near $\mathcal{M}$ and be compatible with gradient-based optimization of $\theta$ as part of a neural underlying ODE. Then, for each time $t$, a $G^+(\boldsymbol{W}_s)$ matrix is employed to obtain the trajectories of dynamic causal structures belonging to $\mathcal{M}$,

$$\boldsymbol{W}_t = \boldsymbol{W}_0 + \int_0^t f_\theta(\boldsymbol{W}_s, s) - \gamma G^+(\boldsymbol{W}_s) h(\boldsymbol{W}_s)\, ds,\ s \in [0, t] \tag{9}$$

where $G^+(\boldsymbol{W}_s) = G^T(\boldsymbol{W}_s)(G(\boldsymbol{W}_s)G^T(\boldsymbol{W}_s))^{-1}$.

## 3.3 CONSTRAINED NEURAL LATENT ODE FOR DYNAMIC DAGS

When inferring dynamic causal structure, $f_\theta(\boldsymbol{W}_s, s)$ can be expensive to solve directly using standard Dormand–Prince methods (Hartman, 2002), especially in real-world applications. This is because $\boldsymbol{W}_s$ can be very high-dimensional. Fortunately, a variety of works provided both theoretical (Holmes, 2012) and empirical evidence (Noack et al., 2011; Sholokhov et al., 2023) that many dynamical systems evolve on a latent space with lower dimensions. Therefore, we follow the "Encoder-Process-Decoder" (Battaglia et al., 2018) fashion to design the neural latent ODE for the intra-slice structure, as sketched in Figure. 2. We now present the details of each component.

**Encoder.** The goal of the encoder is to efficiently compress the intra-slice structure into a compact, lower-dimensional representation, while preserving essential dynamic characteristics, leveraging the inherent sparsity of the DAG structure. For the initial time step $t_0$, the intra-slice structure $\boldsymbol{W}_0 \in$

$\mathbb{R}^{d \times d}$, derived from the observation $\boldsymbol{X}_0$, represents a high-dimensional state, especially when $d$ is sufficiently large. To represent the causal properties of each variable more explicitly, we transform the initial intra-slice structure $\boldsymbol{W}_0$ into $\boldsymbol{S}_0 \in \mathbb{R}^{d \times 2d}$,

$$\boldsymbol{S}_0 = \text{CONCAT}[\boldsymbol{W}_0, \boldsymbol{W}_0^T]. \tag{10}$$

Note that the $i$-th row of $\boldsymbol{W}_0$ indicates which variables depend on $x_i$ at the initial time 0, while the $i$-

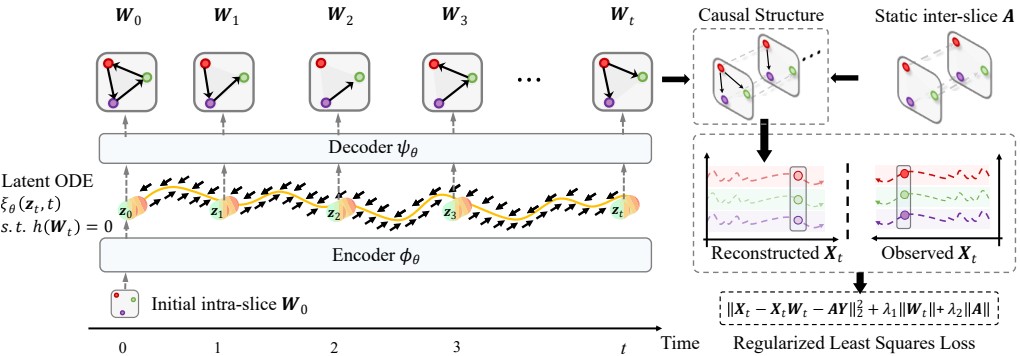

Figure 2: Constrained Neural Latent ODE for constructing causal structures from hidden states.

th column shows which variables $x_i$ depends on. Therefore, the $i$-th row of $\boldsymbol{S}_0$ reflects the variables that $x_i$ is linked to in the DAG within the entire slice. Then, we consider using an encoder $\phi_\theta$ to distill $\boldsymbol{S}_0$ into key features within a $r$-dimensional ($r < d$) latent space. Given the high-dimensional intra-slice structure $\boldsymbol{W}_0$, a vectorized latent representation $\boldsymbol{z}_0 \in \mathbb{R}^{dr}$ is obtained as:

$$\boldsymbol{z}_0 := \text{FLATTEN}(ReLU(\boldsymbol{S}_0 \boldsymbol{P})) \stackrel{\Delta}{=} \phi_\theta(\boldsymbol{S}_0) \tag{11}$$

where $\phi_\theta : \mathbb{R}^{d \times 2d} \mapsto \mathbb{R}^{dr}$ is a mapping parameterized by a neural network, which involves a linear transformation $\boldsymbol{P} \in \mathbb{R}^{2d \times r}$ followed by a ReLU activation function.

**Processor: latent ODE.** In the Processor, the latent state $\boldsymbol{z}_t$ is determined by the mapping $\phi$ from the high-dimensional casual structure matrix $\boldsymbol{W}_t$. Therefore, we can apply the chain rule to derive the dynamics of $\boldsymbol{z}_t$,

$$\frac{d\boldsymbol{z}_t}{dt} = \frac{d\boldsymbol{z}_t}{d\boldsymbol{S}_t}\frac{d\boldsymbol{S}_t}{d\boldsymbol{W}_t}\frac{d\boldsymbol{W}_t}{dt} = \frac{d\boldsymbol{z}_t}{d\boldsymbol{s}_t}\frac{d\boldsymbol{s}_t}{d\boldsymbol{W}_t}f_\theta(\boldsymbol{W}_t, t) \stackrel{\Delta}{=} \xi_\theta(\boldsymbol{z}_t, t). \tag{12}$$

Additionally, since $\boldsymbol{z}_t$ corresponds to $\boldsymbol{W}_t$, and the trajectory of $\boldsymbol{W}_t$ must lie within the DAG manifold, the dynamics of $\boldsymbol{z}_t$ must satisfy some kind of constraint to ensure that the corresponding $\boldsymbol{W}_t$ remains a DAG,

$$h(\psi_\theta(\boldsymbol{z}_t, t)) = 0, \tag{13}$$

where $\psi_\theta$ is the decoder neural network to map the compressed latent state $\boldsymbol{z}_t$ at a given time back to the corresponding intra-slice structure $\boldsymbol{W}_t$ and its details will be described later. Thus, we can integrate such constraint into the latent neural ODE for $\boldsymbol{z}_t$ as follows:

$$\boldsymbol{z}_t = \boldsymbol{z}_0 + \int_0^t \xi_\theta(\boldsymbol{z}_s, s) - \gamma G^+(\psi_\theta(\boldsymbol{z}_s, s))h(\psi_\theta(\boldsymbol{z}_s, s))\, ds,\ s \in [0, t] \tag{14}$$

This Neural ODE aims to learn the dynamic causal structure of time series by exploiting latent states, which can be estimated using a numerical ODE solver,

$$\boldsymbol{z}_1, \dots, \boldsymbol{z}_t = \text{ODESolver}(\xi_\theta, \boldsymbol{z}_0, (t_0, \dots, t)) \tag{15}$$

**Decoder.** In some real-world scenarios, e.g. traffic, the dynamics of the $\boldsymbol{W}_t$ do not only depend on $\boldsymbol{z}_t$, but also explicitly on the time $t$ due to the periodicity of dependency relationship between variables. To deal with this case, we concatenate the latent state $\boldsymbol{z}_t$ with the time variable $t$ to form a temporal-conditioned latent representation $\tilde{\boldsymbol{z}}_t \in \mathbb{R}^{dr+1}$,

$$\tilde{\boldsymbol{z}}_t = \text{CONCAT}[\boldsymbol{z}_t, t] \tag{16}$$

This approach enables the decoder to effectively incorporate time-dependent dynamics in its reconstruction of the intra-slice structure $\boldsymbol{W}_t$. Utilizing the temporal-conditioned latent representation $\tilde{\boldsymbol{z}}_t$, the intra-slice structure $\boldsymbol{W}_t$ can be reconstructed as follows:

$$\boldsymbol{W}_t = \sigma(P_L \cdots \sigma(P_1 \tilde{\boldsymbol{z}}_t)) = \psi_\theta(\boldsymbol{z}_t, t) \tag{17}$$

where $\psi_\theta : \mathbb{R}^{dr+1} \mapsto \mathbb{R}^{d^2}$ is a mapping parameterized by a $L$-layers neural network with nonlinear activation function $\sigma$. We note that if $\boldsymbol{W}_t$ exhibits periodic dynamic behavior, the activation function in $\psi_\theta$ should effectively capture this periodicity. In such cases, we recommend utilizing the SIREN activation function (Sitzmann et al., 2020); otherwise, the SiLU activation function can be employed.

**Optimization problem.** To further enhance the model, we introduce $\ell_1$ penalty terms to enforce sparsity in both $\boldsymbol{W}_t$ and $\boldsymbol{A}$, leading to the following regularized optimization problem:

$$\underset{\theta, \boldsymbol{W}_0, \boldsymbol{A}}{\arg\min} \mathcal{L}(\boldsymbol{A}, \boldsymbol{W}_0, \boldsymbol{W}_t) = \frac{1}{2NT} \sum_{t=0}^{T} \|\boldsymbol{X}_t - \boldsymbol{X}_t \boldsymbol{W}_t(\theta) - \boldsymbol{Y}\boldsymbol{A}\|_2^2 + \lambda_1 \sum_{t=0}^{T} \|\boldsymbol{W}_t(\theta)\|_1 + \lambda_2 \|\boldsymbol{A}\|_1. \tag{18}$$

where $\theta$ represents all the trainable parameters involving in $\boldsymbol{W}_t$, including the encoder $\phi_\theta$, decoder $\psi_\theta$, and vector field function $\xi_\theta$. We use the Adam algorithm (Kingma & Ba, 2015) to solve this objective function.

**Extension of DyCAST.** Our DyCAST modeling framework is flexible, which can be easily incorporated into other casual discovery approaches, particularly those for modeling complex inter-slices temporal causal discovery. When $p = 1$, the summary graph coincides with the inter-slice graph, allowing the use of Granger-causal methods for learning inter-slice structures. For instance, DyCAST can be introduced to CUTS+ (Cheng et al., 2024a), a nonlinear model for inferring inter-slice relationships with summary graphs. Following Section 3.3 as well as the work of NTS-NOTEARS (Sun et al., 2023), the foundational model (Eq.(3)) of DyCAST can be extend as:

$$\boldsymbol{X}_t^i = \boldsymbol{X}_t \boldsymbol{W}_t + \varphi_i(\boldsymbol{X}_{t-p:t-1}^1, \boldsymbol{X}_{t-p:t-1}^2, \cdots, \boldsymbol{X}_{t-p:t-1}^d) + \boldsymbol{Z}_t^i, \tag{19}$$

where the $i$-th neural network $\varphi_i$ predicts the expectation $\mathbb{E}[\boldsymbol{X}_t^i - \boldsymbol{X}_t \boldsymbol{W}_t]$ of the target variable $\boldsymbol{X}_t^i$ at each time step $t$, conditioned on preceding variables. Here, $\varphi_i$ can be implemented as $d$ separate MLPs or LSTMs to ensure disentanglement. Instead of modeling the full-time DAG, we utilize CUTS+ as $\varphi$ to extract a summary graph that captures the causal effects of the lagged version on $\boldsymbol{X}_t^i$.

$$\begin{aligned} \boldsymbol{X}_t &= \boldsymbol{X}_t \boldsymbol{W}_t + \Phi(\boldsymbol{X}_{t-p:t-1}, \boldsymbol{M}) + \boldsymbol{Z}_t^i \\ &= \boldsymbol{X}_t \boldsymbol{W}_t + [..., \text{Linear}_{\varphi_i^2}(\text{MLP}_{\varphi_i^1}(\text{MPGNN}_\nu(\boldsymbol{X}_{t-p:t-1}, h_0^i; m_{:,i})))]^T + \boldsymbol{Z}_t^i \end{aligned} \tag{20}$$

where $h_0^i$ is the initial value of GRU hidden states and irrelevant to $\boldsymbol{X}_{t-p:t-1}$, $\boldsymbol{M}$ is a binary causal matrix, where $m_{:,i} = 1$ denotes $i$-th hidden states Granger cause the prediction. Notably, when the preceding variables span only a single time step, the summary graph is equivalent to the full-time DAG.

## 4 EXPERIMENTAL RESULTS

We evaluate the efficacy of DyCAST through extensive experiments on both synthetic and real-world datasets. For synthetic data, we perform a series of simulation experiments with known ground truth. To demonstrate the broad applicability of our method, we apply it to two real-world datasets: NetSim (Smith et al., 2011) and CausalTime (Cheng et al., 2024b). Additional experiments on variable counts, sequence lengths, and noise robustness, along with ablation studies, are provided in Appendix C. We also demonstrate DyCAST's ability to detect dynamic nonlinear interactions using the Human3.6M dataset, detailed in Appendix E.

**Evaluation Metrics.** We assess the performance of our proposed method for learning dynamic causal structures using three key metrics: 1) F1 score, representing the harmonic mean of precision and recall. 2) Structural Hamming Distance (SHD), which counts discrepancies (e.g., reversed, missing, or redundant edges) between two DAGs; Since the number of potential non-causal relationships vastly outnumbers true causal relationships in real datasets, we also utilize Area Under

the Precision-Recall Curve (AUPRC) and Area Under the ROC Curve (AUROC) to evaluate the effectiveness of DyCAST in identifying genuine causal relationships.

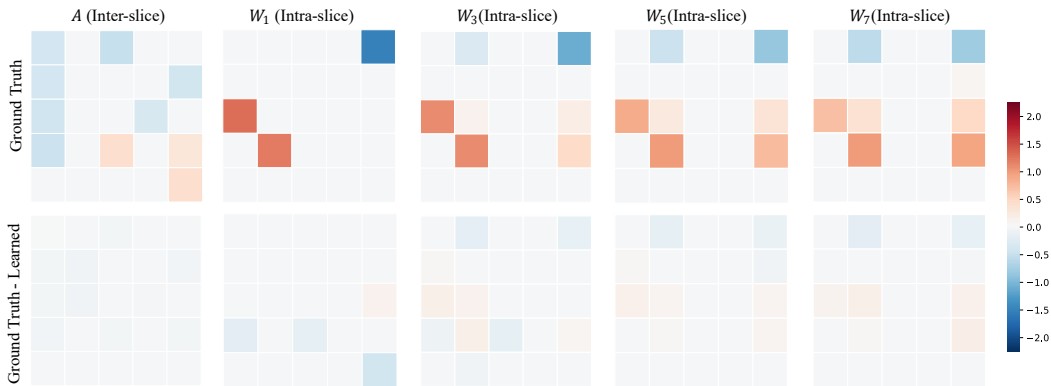

Figure 3: Example results using DyCAST on data with Gaussian noise, consisting of $N = 500$ samples, $T = 7$ time steps, $d = 5$ variables, and $p = 1$ autoregressive term. **The First Row**: The ground truth 1-order dynamic causal structure. **The Second Row**: The difference between ground truth and learned dynamic structures over time.

## 4.1 SYNTHETIC DATASETS

We first follow the procedure outlined in Appendix A.1 to generate data with Gaussian noise, consisting of $N = 500$ samples, $T = 8$ time steps, $d = 5$ variables, and $p = 1$ autoregressive term. The evolution function $F$ is represented by a linear transformation matrix, with both intra- and inter-slice DAGs simulated using an Erdős–Rényi (ER) model (Erdős & Rényi, 1960), where the mean degree is set to 2. We implement DyCAST to this dataset using regularization parameters $\lambda_1 = \lambda_2 = 0.05$ and scalar parameter $\gamma = 1$. In Figure. 3, we demonstrate the performance of DyCAST on this synthetic dataset. We can clearly see that the estimated weights closely match the ground truth for both $W_t$ and $A$. Furthermore, the temporal dynamics of $W_t$ are effectively captured.

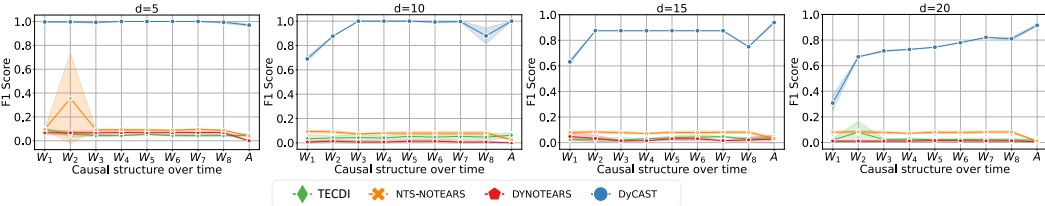

Figure 4: F1 scores for different temporal causal discovery algorithms and different numbers of variables $d \in \{5, 10, 15, 20\}$ on synthetic dataset with dynamic causal structure. Each panel contains results for both intra- and inter-slice structures. Every marker corresponds to the mean performance across 4 algorithm runs, each on a different simulated dataset.

**Performance on time series with dynamic causal structure.** We compare DyCAST against several causal discovery baselines, including DYNOTEARS (Pamfil et al., 2020), NTS-NOTEARS (Sun et al., 2023) and TECDI (Li et al., 2023), on a synthetic dataset with variables dimension $d \in \{5, 10, 15, 20\}$, $N = 500$ samples, $T = 8$ time steps, and $p = 1$ autoregressive term. The results are presented in the Figure 4. The vertical axis indicates the performance of each algorithm measured by the F1 score, computed separately for intra- and inter-slice structures. It is clear that DyCAST outperforms the others, consistently attaining F1 scores near 1 over time. The selection of hyperparameter values for the four algorithms is discussed in Appendix B.

**Performance on time series with static causal structure.** Here we show that DyCAST can also naturally adapt to time series with a static causal structure. We apply it to a synthetic dataset similar to that in Pamfil et al. (2020), where only $W_t$ remains fixed over time. As anticipated, DyCAST

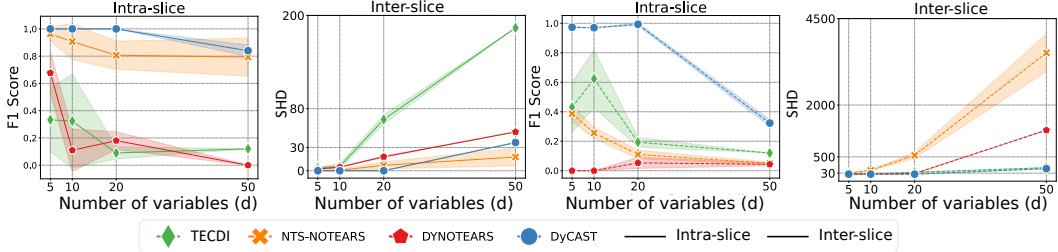

Figure 5: Left two panels: the intra-slice F1 scores and SHD metrics for various temporal causal discovery algorithms across different variable counts $d \in \{5, 10, 20, 50\}$ on synthetic dataset with static causal structure. Right two panels: the inter-slice F1 scores and SHD metrics for various temporal causal discovery algorithms.

Table 1: Ablation study on synthetic datasets with $d = 5$.

| Structure | | DYNOTEARS | | | w/o $S_0$ | | | w/o Latent state | | | **DyCAST** | | |
|---|---|---|---|---|---|---|---|---|---|---|---|---|---|
| | | TPR | SHD | F1 | TPR | SHD | F1 | TPR | SHD | F1 | TPR | SHD | F1 |
| DYNAMIC DATASET | $W_2$ | 0.40 | 3 | 0.67 | 0.86 | 3 | 0.75 | 1.00 | 2 | 0.67 | **1.00** | **0** | **1.00** |
| | $W_4$ | 0.40 | 3 | 0.69 | 0.86 | 3 | 0.75 | 1.00 | 2 | 0.78 | **1.00** | **0** | **1.00** |
| | $W_6$ | 0.80 | 3 | 0.68 | 0.86 | 4 | 0.75 | 1.00 | 2 | 0.82 | **1.00** | **0** | **1.00** |
| | $W_8$ | 0.40 | 3 | 0.68 | 1.00 | 6 | 0.59 | 1.00 | 2 | 0.93 | **1.00** | **0** | **1.00** |
| | $A$ | 0.20 | 3 | 0.00 | 1.00 | 12 | 0.37 | 1.00 | 3 | 0.86 | **1.00** | 1 | 0.97 |
| STATIC DATASET | $W$ | 0.80 | 1 | 0.89 | 1.00 | 7 | 0.53 | 1.00 | 1 | 0.89 | **1.00** | **0** | **1.00** |
| | $A$ | 0.00 | 5 | 0.00 | 1.00 | 7 | 0.67 | 1.00 | 4 | 0.82 | **1.00** | **0** | **1.00** |

The best results are in **bold**.

emerges as the best-performing algorithm in Figure 5, demonstrating high F1 scores and low SHD values. The second-best algorithm is NTS-NOTEARS. However, its performance tends to degrade as we add the number of variables. Moreover, DyCAST reduces the running time by more than 20% across various variable dimensions. See Appendix C.6 for more details.

**Ablation study.** We conduct an ablation study by evaluating DyCAST without latent states, as shown in Table 1. Specifically, we run DYNOTEARS, DyCAST, DyCAST w/o $S_0$ (which means $z_0 := \text{FLATTEN}(ReLU(W_0 P))$) and DyCAST w/o Latent states (which means $z_0 := \text{FLATTEN}(W_0)$) on two datasets: a 5-variable synthetic dataset with dynamic causal structure, and a 5-variable synthetic dataset with static causal structure. The results from these experiments provide insights into the contribution of latent states and $S_0$ to the overall performance of DyCAST. The gray shadows in tables that are highlighted in all tables indicate our choice for our method. We find that $S_0$ plays an important role in improving the performance, while latent states can further enhance the effect. See Appendix D for real world datasets ablation study.

## 4.2 NETSIM DATASETS

**Baseline methods.** For NetSim datasets, we compare DyCAST with GC (Granger, 1969), DYNOTEARS (Pamfil et al., 2020), NTS-NOTEARS (Sun et al., 2023), PCMCI (Peters et al., 2013), NGC (Tank et al., 2022), CUTS+ (Cheng et al., 2024a), LCCM (Brouwer et al., 2021), eSRU (Khanna & Tan, 2020), TECDI (Li et al., 2023).

NetSim is an fMRI dataset aimed at learning gradual changes in brain networks to enhance the understanding of brain functions and intelligence (Smith et al., 2011). It contains 28 simulation datasets, from which we select 17 simulations with the same sequence length, each featuring 50 independent time series recordings for $d = \{5, 10, 15\}$ nodes over 200 time steps. A detailed comparison of AUPRC is shown in Table 2, see Appendix C.10 for AUROC results and Appendix D for ablation study. Given that the NetSim dataset explores brain network causal relationships, whose structure does not fully adhere to the DAG constraint. We also evaluated a variant of DyCAST without the DAG manifold constraint, referred to as DyCAST (No DAG). While TECDI performs better than other variants of NOTEARS (DYNOTEARS, NTS-NOTEARS), DyCAST outperforms

Table 2: Performance on NetSim Dataset Under AUPRC.

| Metric | DATASET | GC | DYNOTEARS | NTS-NOTEARS | PCMCI | NGC | CUTS+ | LCCM | eSRU | TECDI | DyCAST | DyCAST (Not DAG) | DyCAST-CUTS+ |
|---|---|---|---|---|---|---|---|---|---|---|---|---|---|
| AUPRC | Sim1 | 0.40±0.08 | 0.41±0.08 | 0.41±0.06 | 0.39±0.09 | 0.42±0.15 | 0.85±0.11 | 0.71±0.14 | 0.40±0.14 | 0.67±0.03 | 0.90±0.13 | 0.79±0.05 | 0.92±0.03 |
| | Sim2 | 0.32±0.12 | 0.33±0.12 | 0.24±0.04 | 0.29±0.11 | 0.29±0.11 | 0.79±0.12 | 0.82±0.12 | 0.27±0.11 | 0.79±0.04 | 0.91±0.11 | 0.85±0.04 | 0.89±0.12 |
| | Sim3 | 0.29±0.14 | 0.32±0.13 | 0.16±0.02 | 0.26±0.12 | 0.26±0.12 | 0.77±0.10 | 0.78±0.08 | 0.23±0.12 | 0.73±0.04 | 0.84±0.04 | 0.87±0.01 | 0.85±0.04 |
| | Sim8 | 0.38±0.11 | 0.36±0.08 | 0.42±0.04 | 0.36±0.10 | 0.40±0.14 | 0.85±0.09 | 0.84±0.04 | 0.39±0.14 | 0.58±0.10 | 0.89±0.14 | 0.76±0.08 | 0.90±0.08 |
| | Sim10 | 0.39±0.12 | 0.38±0.10 | 0.46±0.05 | 0.40±0.12 | 0.42±0.16 | 0.71±0.02 | 0.68±0.11 | 0.42±0.15 | 0.71±0.05 | 0.90±0.14 | 0.77±0.05 | 0.91±0.10 |
| | Sim11 | 0.26±0.06 | 0.26±0.04 | 0.21±0.02 | 0.25±0.07 | 0.25±0.08 | 0.78±0.04 | 0.70±0.02 | 0.24±0.08 | 0.74±0.02 | 0.84±0.10 | 0.67±0.02 | 0.85±0.08 |
| | Sim12 | 0.33±0.11 | 0.36±0.08 | 0.21±0.02 | 0.29±0.11 | 0.28±0.11 | 0.79±0.02 | 0.73±0.02 | 0.26±0.11 | 0.79±0.02 | 0.92±0.08 | 0.69±0.02 | 0.90±0.03 |
| | Sim13 | 0.48±0.07 | 0.47±0.05 | 0.53±0.05 | 0.47±0.10 | 0.47±0.10 | 0.83±0.01 | 0.85±0.15 | 0.47±0.11 | 0.68±0.02 | 0.81±0.10 | 0.84±0.11 | 0.80±0.12 |
| | Sim14 | 0.41±0.09 | 0.41±0.08 | 0.42±0.02 | 0.38±0.09 | 0.41±0.13 | 0.77±0.04 | 0.73±0.10 | 0.39±0.13 | 0.67±0.04 | 0.87±0.13 | 0.73±0.07 | 0.80±0.07 |
| | Sim15 | 0.40±0.09 | 0.38±0.07 | 0.42±0.03 | 0.41±0.10 | 0.47±0.20 | 0.82±0.01 | 0.81±0.06 | 0.44±0.19 | 0.72±0.04 | 1.00±0.20 | 0.74±0.08 | 0.98±0.02 |
| | Sim16 | 0.45±0.07 | 0.44±0.05 | 0.50±0.05 | 0.44±0.06 | 0.46±0.10 | 0.80±0.09 | 0.79±0.04 | 0.45±0.10 | 0.64±0.05 | 0.81±0.10 | 0.77±0.08 | 0.82±0.08 |
| | Sim17 | 0.36±0.10 | 0.39±0.09 | 0.22±0.02 | 0.35±0.10 | 0.40±0.19 | 0.77±0.15 | 0.84±0.10 | 0.35±0.19 | 0.86±0.02 | 0.89±0.19 | 0.82±0.12 | 0.90±0.13 |
| | Sim18 | 0.42±0.12 | 0.42±0.07 | 0.41±0.01 | 0.40±0.11 | 0.42±0.16 | 0.85±0.02 | 0.83±0.06 | 0.40±0.15 | 0.68±0.05 | 0.88±0.16 | 0.72±0.15 | 0.86±0.08 |
| | Sim21 | 0.41±0.08 | 0.42±0.08 | 0.43±0.06 | 0.38±0.09 | 0.41±0.14 | 0.85±0.09 | 0.87±0.08 | 0.39±0.14 | 0.68±0.03 | 0.92±0.14 | 0.76±0.04 | 0.92±0.09 |
| | Sim22 | 0.38±0.08 | 0.38±0.06 | 0.45±0.04 | 0.37±0.08 | 0.35±0.09 | 0.86±0.10 | 0.86±0.09 | 0.34±0.09 | 0.83±0.04 | 0.80±0.09 | 0.76±0.03 | 0.88±0.12 |
| | Sim23 | 0.40±0.12 | 0.35±0.06 | 0.41±0.03 | 0.41±0.14 | 0.45±0.20 | 0.79±0.06 | 0.79±0.07 | 0.42±0.19 | 0.78±0.02 | 0.73±0.20 | 0.76±0.14 | 0.81±0.06 |
| | Sim24 | 0.34±0.10 | 0.31±0.07 | 0.47±0.07 | 0.35±0.11 | 0.34±0.11 | 0.56±0.08 | 0.60±0.03 | 0.34±0.11 | 0.79±0.02 | 0.61±0.11 | 0.75±0.05 | 0.76±0.07 |
| | Average | 0.38±0.10 | 0.38±0.08 | 0.37±0.04 | 0.36±0.10 | 0.38±0.13 | 0.79±0.07 | 0.82±0.08 | 0.36±0.14 | 0.73±0.04 | 0.85±0.13 | 0.77±0.07 | 0.87±0.08 |

The best results are in **bold** and the second best are underlined.

TECDI on 82.4% of datasets. Overall, DyCAST achieves an average AUPRC increase of 16.43%, demonstrating adaptability to low-resolution datasets with significant time intervals.

Additionally, we found that DyCAST (Not DAG) can obtain an AUROC value close to 1.0 on some sub-datasets of the NetSim dataset. This is because after eliminating the DAG constraint, DyCAST can identify the influence relationship between variables, but it is not accurate in identifying the direction of the influence. Therefore, the AUPRC of DyCAST (Not DAG) is relatively low. Moreover, due to the highly nonlinear relationships between variables in the NetSim dataset, nonlinear causal modeling methods like CUTS+ and LCCM demonstrate higher accuracy compared to linear methods. Similarly, the DyCAST variant integrated with CUTS+ achieves an average 2.4% higher accuracy than the vanilla DyCAST.

## 4.3 CAUSALTIME DATASETS

**Baseline methods.** For CausalTime datasets, we compare DyCAST with GC (Granger, 1969), SVAR, NTS-NOTEARS (Sun et al., 2023), PCMCI (Peters et al., 2013), Rhino (Gong et al., 2023), CUTS (Cheng et al., 2023), CUTS+ (Cheng et al., 2024a), NGC (Tank et al., 2022), NGM (Bellot et al., 2022), LCCM (Brouwer et al., 2021), eSRU (Khanna & Tan, 2020), SGCL (Xu et al., 2019), TCDF (Nauta et al., 2019), TECDI (Li et al., 2023).

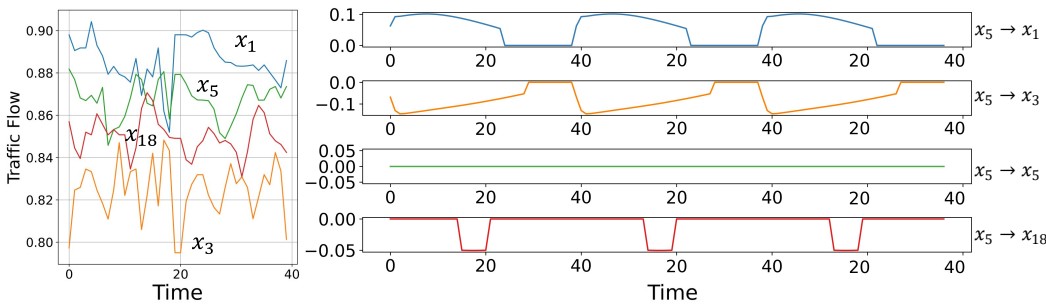

Figure 6: Visualization of the dynamic causal structure on Traffic. Left: the original traffic time series. Right: the periodic changes of some part edge links in intra-slice structures discovered by DyCAST.

We evaluate DyCAST on the CausalTime dataset, a real-world temporal causal discovery benchmark (Cheng et al., 2024b). CausalTime includes three distinct scenarios: weather (AQI subset), traffic (Traffic subset), and healthcare (Medical subset). The Traffic and Medical subsets each consist of $d = 20$ variables, while the AQI subset contains $d = 36$ variables. Table 3 shows the performance comparison on temporal causal discovery, see Appendix D for ablation study. We can observe that: (1) Among all the compared methods, DyCAST performs best in the Traffic subset and second-best in Medical. One possible reason is that DyCAST is more effective on datasets with rapidly

changing causal structures. See Figure 6 for more details. (2) Our DyCAST achieves the best average performance, likely due to the explicit incorporation of both instantaneous and lagged causality information, which enhances its ability to capture complex causal dynamics.

Table 3: Performance on CausalTime Dataset Under AUROC and AUPRC. DyCAST-CUTS combines DyCAST for dynamic intra-slice discovery and CUTS+ for nonlinear inter-slice discovery, achieving the best performance.

| Methods | AUROC | | | AUPRC | | |
|---|---|---|---|---|---|---|
| | AQI | Traffic | Medical | AQI | Traffic | Medical |
| GC | $0.45_{\pm0.04}$ | $0.42_{\pm0.03}$ | $0.57_{\pm0.03}$ | $0.63_{\pm0.02}$ | $0.28_{\pm0.00}$ | $0.42_{\pm0.03}$ |
| SVAR | $0.62_{\pm0.04}$ | $0.63_{\pm0.00}$ | $0.71_{\pm0.02}$ | $0.79_{\pm0.02}$ | $0.58_{\pm0.00}$ | $0.68_{\pm0.04}$ |
| NTS-NOTEARS | $0.57_{\pm0.02}$ | $0.63_{\pm0.03}$ | $0.71_{\pm0.02}$ | $0.71_{\pm0.02}$ | $0.58_{\pm0.05}$ | $0.46_{\pm0.02}$ |
| PCMCI | $0.53_{\pm0.07}$ | $0.54_{\pm0.07}$ | $0.70_{\pm0.01}$ | $0.67_{\pm0.04}$ | $0.35_{\pm0.06}$ | $0.51_{\pm0.02}$ |
| Rhino | $0.67_{\pm0.10}$ | $0.63_{\pm0.02}$ | $0.65_{\pm0.02}$ | $0.76_{\pm0.08}$ | $0.38_{\pm0.01}$ | $0.49_{\pm0.03}$ |
| CUTS | $0.60_{\pm0.00}$ | $0.62_{\pm0.02}$ | $0.37_{\pm0.03}$ | $0.51_{\pm0.04}$ | $0.15_{\pm0.02}$ | $0.15_{\pm0.00}$ |
| CUTS+ | $\underline{0.89}_{\pm0.02}$ | $0.62_{\pm0.07}$ | $\underline{0.82}_{\pm0.02}$ | $0.80_{\pm0.08}$ | $0.64_{\pm0.12}$ | $0.55_{\pm0.13}$ |
| NGC | $0.72_{\pm0.01}$ | $0.60_{\pm0.01}$ | $0.57_{\pm0.01}$ | $0.72_{\pm0.01}$ | $0.36_{\pm0.05}$ | $0.46_{\pm0.01}$ |
| NGM | $0.67_{\pm0.02}$ | $0.47_{\pm0.01}$ | $0.56_{\pm0.02}$ | $0.48_{\pm0.02}$ | $0.28_{\pm0.01}$ | $0.47_{\pm0.02}$ |
| LCCM | $0.86_{\pm0.07}$ | $0.55_{\pm0.03}$ | $0.80_{\pm0.02}$ | $\mathbf{0.93}_{\pm0.02}$ | $0.59_{\pm0.05}$ | $\underline{0.76}_{\pm0.02}$ |
| eSRU | $0.83_{\pm0.03}$ | $0.60_{\pm0.02}$ | $0.76_{\pm0.04}$ | $0.72_{\pm0.03}$ | $0.49_{\pm0.03}$ | $0.74_{\pm0.06}$ |
| SCGL | $0.49_{\pm0.05}$ | $0.59_{\pm0.06}$ | $0.50_{\pm0.02}$ | $0.36_{\pm0.03}$ | $0.45_{\pm0.03}$ | $0.48_{\pm0.02}$ |
| TCDF | $0.41_{\pm0.02}$ | $0.50_{\pm0.00}$ | $0.63_{\pm0.04}$ | $0.65_{\pm0.01}$ | $0.36_{\pm0.00}$ | $0.55_{\pm0.03}$ |
| TECDI | $0.56_{\pm0.02}$ | $0.60_{\pm0.00}$ | $0.63_{\pm0.02}$ | $0.65_{\pm0.00}$ | $0.63_{\pm0.01}$ | $0.49_{\pm0.01}$ |
| **DyCAST (Not DAG)** | $0.85_{\pm0.03}$ | $\underline{0.68}_{\pm0.05}$ | $0.74_{\pm0.01}$ | $0.70_{\pm0.03}$ | $0.60_{\pm0.02}$ | $0.69_{\pm0.03}$ |
| **DyCAST** | $0.85_{\pm0.02}$ | $0.63_{\pm0.01}$ | $0.81_{\pm0.03}$ | $\underline{0.82}_{\pm0.00}$ | $\underline{0.70}_{\pm0.01}$ | $0.74_{\pm0.03}$ |
| **DyCAST-CUTS+** | $\mathbf{0.91}_{\pm0.02}$ | $\mathbf{0.65}_{\pm0.01}$ | $\mathbf{0.84}_{\pm0.01}$ | $\mathbf{0.93}_{\pm0.00}$ | $\mathbf{0.73}_{\pm0.01}$ | $\mathbf{0.77}_{\pm0.03}$ |

The best results are in **bold** and the second best are underlined.

Since our method focuses primarily on the dynamic intra-slice structure, we do not heavily address the inter-slice structure. To comprehensively account for both aspects, we combine DyCAST with other methods that emphasize inter-slice structure. As expected, DyCAST-CUTS+ achieves the best performance across all three datasets.

**Visualization of the dynamic causal structure.** Figure 6 provides visualizations of the dynamic causal structure learned by DyCAST, using the Traffic subset as an illustrative example. The left panel shows the changes in the original traffic time series within a day. The right panel shows the periodic changes in the intra-slice edge links between $x_5$ and other variables found by DyCAST. We use blue to represent $x_1$, orange to represent $x_3$, green to represent $x_5$, and red to represent $x_{18}$. We can observe that within a period of 1 day, the causal relationship between intersection $x_5$ and other intersections $x_1, x_3, x_5, x_{18}$ shows obvious dynamics and periodicity.

## 5 CONCLUSION

In this work, we introduce DyCAST, a novel constrained latent Neural ODE-based temporal causal discovery framework, specifically designed to learn dynamic causal structures from time series data. We conduct extensive experiments on dynamic synthetic, static synthetic, and real-world time series datasets, demonstrating DyCAST's superior performance in recovering causal structures across all scenarios. In the future, we plan to explore new architectural designs for neural networks in the latent dynamics model constrained by the DAG manifold. Another direction involves integrating both intra-slice and inter-slice latent dynamics within a unified neural network framework.

## ACKNOWLEDGEMENT

This work was supported by the Beijing Natural Science Foundation under Grant (No.L231005), and by the National Key Research and Development Program of China under Grant (No.2024YFB3312200).

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

APPENDIX

## A  IMPLEMENTATION DETAILS

In this section, we first provide the specific steps and details of synthetic data generation. Secondly, we introduce the structure of each component of DyCAST in its specific implementation.

### A.1  DATA GENERATION PROCESS

We provide more details about the data generation process that we use in our numerical experiments form Section 4.1.

**Dynamic intra-slice model.** As in Zheng et al. (2018), we use the Erdős–Rényi (ER) model to generate the initial intra-slice DAG $W_0$. Similarly, the intra-slice DAG $W_T$ at the final time step is generated using the same approach as the initial structure. To assign weights to the DAG, we sample uniformly from the intervals $[-2.0, -0.5] \cup [0.5, 2.0]$. Given the assumption of smooth causal transitions, we generate the intra-slice DAGs for the remaining $t - 1$ transition times using linear interpolation. At the $i$-th time step, the interpolated matrix $W_i$ is defined as follows:

$$W_i = (1 - t_i) \cdot W_0 + t_i \cdot W_t \tag{21}$$

where $t_i = \frac{i}{T-1}$, with $i$ representing the interpolation step. As $i$ increases, the interpolation coefficient $t_i$ linearly progresses from 0 to 1, smoothly transitioning from $W_0$ to $W_T$.

**Static inter-slice model.** We employ a directed Erdős–Rényi (ER) model to generate inter-slice DAGs. Specifically, we sample entries of the binary adjacency matrix $A^{bin}$ using independent and identically distributed (i.i.d.) Bernoulli trials with probabilities of $k/d$. Given the binary inter-slice adjacency matrix $A^{bin}$, we sample edge weights uniformly from a specified interval that depends on the parameter $p$. More precisely, edge weights from slice $t - p$ to slice $t$ are sampled from the intervals $[-0.5\alpha, -0.3\alpha] \cup [0.3\alpha, 0.5\alpha]$, where $\alpha = 1/\eta^{p-1}$ and $\eta = 1.5$. This weight decay parameter $\eta$ effectively reduces the influence of variables that are farther back in time relative to the current time slice.

Once we have $W_t$ and $A$, we utilize the SEM from Eq. (1) to generate a data matrix $X$ of size $n \times T \times d$. The noise term $Z_t$ in Eq. (1) consists of independent and identically distributed (i.i.d.) random variables.

### A.2  DYCAST IMPLEMENTATION DETAILS

**The encoder part:** Similar to Neural LAD, we solve the constrained Neural latent ODE with defined $\phi_\theta$, which is parameterized as a single layer MLP with ReLU.

**The vector field part:** The vector field of DyCAST is constructed by stacking multiple linear layers, each followed by a tanh activation function. This structure enables the model to capture complex dynamics of hidden states.

**The decoder part:** we stack multiple linear layers with a non-linear activation function to parameterize the neural network $\psi_\theta$. Specifically, we configure the first layer as a multi-layer perceptron (MLP) with an input dimension of $r \times d$ and an output dimension corresponding to the hidden layer. The final layer has an input dimension equal to the hidden layer and an output dimension of $d \times d$ to generate the intra-slice structure. Typically, the decoder employs the SiLU activation function. However, if the intra-slice structure is known to exhibit periodicity, we select the SIREN activation function instead.

## B  EXPERIMENTS DETAILS

We show the detailed settings of hyper-parameters including learning rate, hidden dimensions, and stable matrix scale factor. We run all experiments on an NVIDIA GeForce RTX 4090 GPU. It is worth that the DyCAST converges faster than the other variants NOTEARS, so it achieves better performance earlier than baselines.

Table 4: Detailed hyper-parameter settings of all networks on all datasets.

| Type | Dataset | Lags | lr | $r$ | $\gamma$ | Activation |
|---|---|---|---|---|---|---|
| DYNAMIC | d=5 | 1 | 1.00E-03 | 4 | 1 | SiLU |
| | d=10 | 1 | 1.00E-03 | 8 | 1 | SiLU |
| | d=15 | 1 | 1.00E-03 | 12 | 1 | SiLU |
| | d=20 | 1 | 1.00E-03 | 16 | 1 | SiLU |
| | d=50 | 1 | 1.00E-03 | 40 | 1 | SiLU |
| | d=100 | 1 | 1.00E-03 | 80 | 0.1 | SiLU |
| | d=200 | 1 | 1.00E-03 | 180 | 0.01 | SiLU |
| | d=300 | 1 | 1.00E-03 | 240 | 0.01 | SiLU |
| STATIC | d=5 | 1 | 1.00E-03 | 4 | 1 | SiLU |
| | d=10 | 1 | 1.00E-03 | 8 | 1 | SiLU |
| | d=20 | 1 | 1.00E-03 | 16 | 1 | SiLU |
| | d=50 | 1 | 1.00E-03 | 40 | 1 | SiLU |
| NETSIM | d=5 | 1 | 1.00E-03 | 4 | 1 | SiLU |
| | d=10 | 1 | 1.00E-03 | 8 | 1 | SiLU |
| | d=15 | 1 | 1.00E-03 | 12 | 1 | SiLU |
| CAUSALTIME | d=20 (Traffic) | 1 | 1.00E-03 | 16 | 1 | Siren |
| | d=36 (AQI) | 1 | 1.00E-03 | 30 | 1 | SiLU |
| | d=20 (Medical) | 1 | 1.00E-03 | 16 | 1 | SiLU |
| HUMAN3.6M | d=16 | 1 | 1.00E-03 | 16 | 1 | SiLU |

## C   ADDITIONAL QUANTITATIVE RESULTS

In this section, we add several quantitative experiments, including analyses on variable counts, sample sizes, sequence lengths, the rank of causal structures, unknown autoregressive order, noise robustness, and running times.

### C.1   LARGE-SCALE VARIABLE OF DATASETS.

This section provides additional results of DyCAST on the large-scale variable datasets for Section 4.1, as shown in Table 5. We run DyCAST and other baselines on the synthetic datasets with variables dimension $d \in \{50, 100, 200, 300\}$, $N = 500$ samples, $T = 8$ time steps, and $p = 1$ autoregressive term.

Table 5:  Performance on large-scale variable datasets.

| Methods | $d = 50$ | | | $d = 100$ | | | $d = 200$ | | | $d = 300$ | | |
|---|---|---|---|---|---|---|---|---|---|---|---|---|
| | TPR | SHD | F1 | TPR | SHD | F1 | TPR | SHD | F1 | TPR | SHD | F1 |
| DYNOTEARS | $1.00_{\pm 0.00}$ | $2500_{\pm 0.00}$ | $0.00_{\pm 0.00}$ | - | - | - | - | - | - | - | - | - |
| NTS-NOTEARS | $1.00_{\pm 0.00}$ | $2500_{\pm 0.00}$ | $0.00_{\pm 0.00}$ | - | - | - | - | - | - | - | - | - |
| TECDI | $1.00_{\pm 0.00}$ | $2500_{\pm 0.00}$ | $0.00_{\pm 0.00}$ | - | - | - | - | - | - | - | - | - |
| **DyCAST** | $0.89_{\pm 0.02}$ | $9.91_{\pm 3.14}$ | $0.73_{\pm 0.05}$ | $0.87_{\pm 0.12}$ | $16_{\pm 4.17}$ | $0.68_{\pm 0.08}$ | $0.61_{\pm 0.10}$ | $18.89_{\pm 5.53}$ | $0.67_{\pm 0.04}$ | $0.43_{\pm 0.04}$ | $49.11_{\pm 7.14}$ | $0.56_{\pm 0.03}$ |

We can see that DYNOTEARS, NTS-NOTEARS, and TECDI fail to converge when the number of variables reaches 50, incorrectly inferring causal relationships for all edges. In contrast, DyCAST achieves an average F1 score of 0.73 under the same conditions. Scaling up to 300 variables, these baselines remain non-convergent, while DyCAST maintains an average F1 score of 0.56 and a low SHD ( $49.11_{\pm 7.14}$ ), demonstrating robustness in handling large-scale causal structures.

### C.2   SMALL SAMPLE SIZES DATASETS.

This section provides additional results of DyCAST on small sample sizes for Section 4.1, as shown in Figure 7. We run DyCAST and other baselines on the synthetic datasets with sample size $n \in \{20, 50, 100, 200, 300, 400\}$, $d = 20$ variables dimension, $T = 8$ time steps, and $p = 1$ autoregressive term.

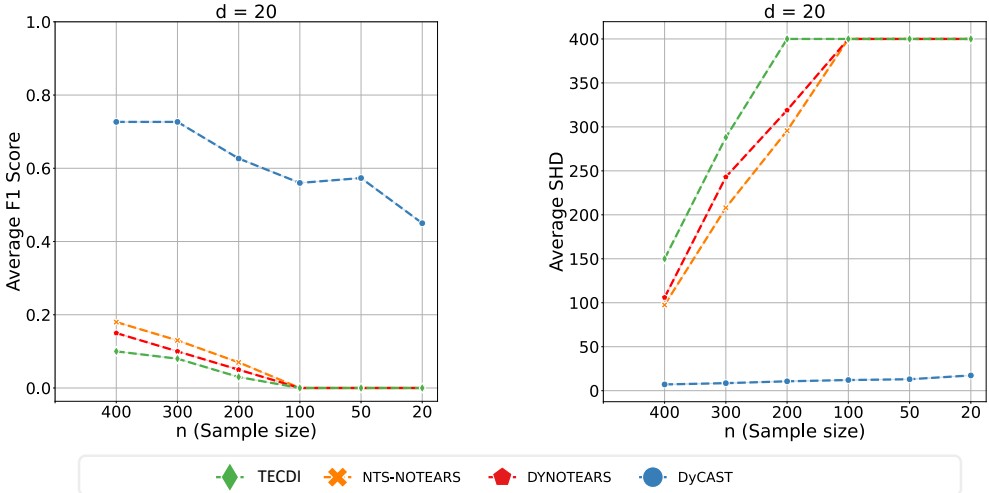

Figure 7: Performance on small sample sizes datasets. Left panel: Average F1 scores for different temporal causal discovery algorithms and different sample sizes $n \in \{400, 300, 200, 100, 50, 20\}$ on synthetic dataset with dynamic causal structure. Right panel: Average SHD for different temporal causal discovery algorithms and different sample sizes on the synthetic dataset with dynamic causal structure.

As sample size decreases, accuracy drops across all methods, but DYNOTEARS, NTS-NOTEARS, and TECDI fail to converge with fewer than 100 samples. In contrast, DyCAST achieves an average F1 score of 0.41 with just 20 samples, maintaining a low average SHD (20) per time step, highlighting its data efficiency compared to other methods.

### C.3 DIFFERENT TIME STEPS OF DATASETS.

This section provides additional results of DyCAST on different time steps of datasets for Section 4.1, as shown in Figure 8. We run DyCAST and other baselines on the synthetic datasets with time steps $T \in \{8, 16, 32, 64, 128\}$, $d = 10$ variables dimension, $N = 500$ samples, and $p = 1$ autoregressive term.

As time steps increase, the complexity of intra-slice dynamics grows, leading to a gradual decline in DyCAST's accuracy. Nonetheless, even with ultra-long time steps ($T = 128$), DyCAST achieves an average F1 score of 0.5, while avoiding the SHD surge observed in baseline algorithms.

### C.4 DIFFERENT NOISE STRENGTH AND TYPES OF DATASETS.

This section provides additional results of DyCAST on noise strength and types of datasets for Section 4.1, as shown in Figure 9.

As noise increases, the performance of causal graphs estimated by DyCAST declines. Notably, the intra-slice matrix of the first time step is the most sensitive to noise. While the accuracy of intra-slice matrices for subsequent time steps also decreases, the overall performance remains high. This is because the first time step relies solely on initial conditions, lacking prior variables to aid in estimating its dynamics. Additionally, we also run DyCAST on datasets with various types of noise, and the results demonstrated that DyCAST performs robustly across different noise conditions, as shown in the right two panels of Figure 9.

### C.5 AUTOREGRESSIVE ORDER.

This section provides additional results of DyCAST on the correct value of the autoregressive order $p$ is unknown, as shown in Figure 10 and 12. We first follow the procedure outlined in Appendix A.1 to generate data with Gaussian noise, consisting of $N = 500$ samples, $T = 8$ time steps, $d = 10$ variables, and $p_{true} = 1$ autoregressive term, as shown in the first row of Figure 10. We then run

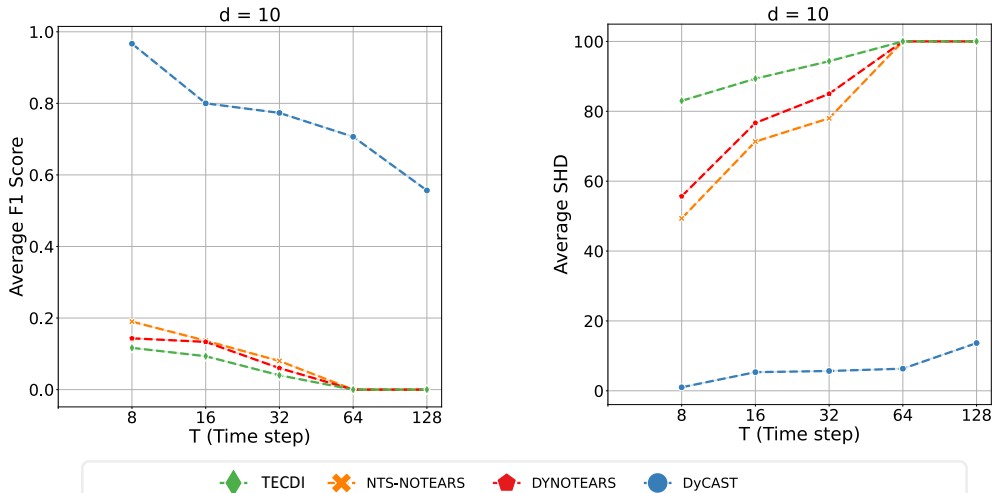

Figure 8: Performance on small sample sizes datasets. Left panel: Average F1 scores for different temporal causal discovery algorithms and time steps $T \in \{8, 16, 32, 64, 128\}$ on synthetic dataset with dynamic causal structure. Right panel: Average SHD for different temporal causal discovery algorithms and different time steps on the synthetic dataset with dynamic causal structure.

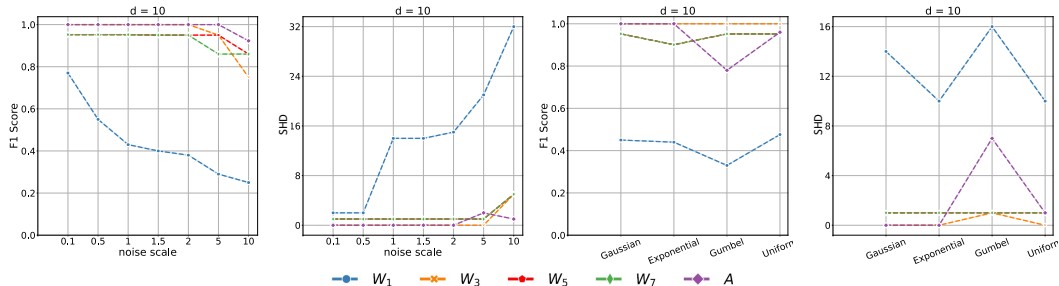

Figure 9: Left two panels: F1 scores and SHD for different temporal causal discovery algorithms and different noise strengths $\{0.1, 0.5, 1, 1.5, 2, 5, 10\}$ of the synthetic dataset with dynamic causal structure. Right two panels: F1 scores and SHD for different temporal causal discovery algorithms and different noise types {Gaussian, Exponential, Gumbel, Uniform} of the synthetic dataset with dynamic causal structure. Each panel contains results for both intra- and inter-slice structures. Every marker corresponds to the mean performance across 4 algorithm runs, each on a different simulated dataset.

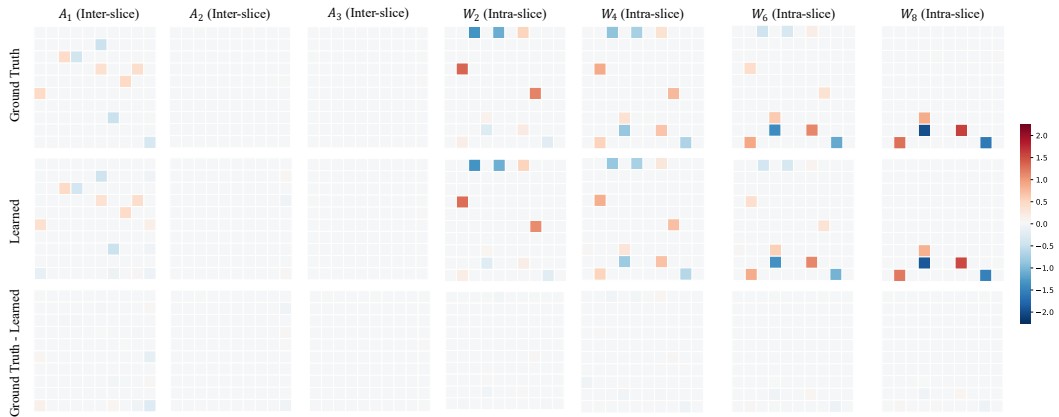

Figure 10: Example results using DyCAST on data with Gaussian noise, consisting of $N = 500$ samples, $T = 8$ time steps, $d = 10$ variables, and $p = 1$ autoregressive term, while the $\hat{p}$=3. **The First Row**: The ground truth 1-order dynamic causal structure. **The Second Row**: The learned 3-order dynamic causal structure. **The third Row**: The difference between ground truth and learned dynamic structures over time.

DyCAST to this dataset using an estimate autoregressive term $p = 3$, regularization parameters $\lambda_1 = \lambda_2 = 0.05$ and scalar parameter $\gamma = 1$. We can clearly see that the estimated weights closely match the ground truth for both $\boldsymbol{W}_t$ and $\boldsymbol{A}$, the estimated inter-slice matrices reveal that only the 2nd-order matrix $\boldsymbol{A}_2$ contains entries with very small values, while the 3rd-order matrix $\boldsymbol{A}_3$ aligns closely with the ground truth. Thus, we suggest that when $p$ is unknown, opting for a slightly larger value is preferable.

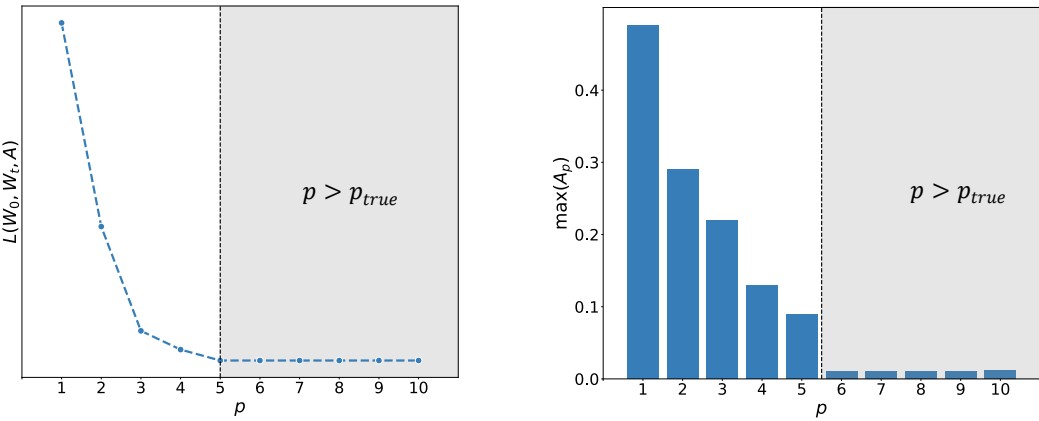

Figure 11: Left panel: Objective value as a function of $p$. Right panel: Largest absolute value in $\boldsymbol{A}_p$ as a function of $p$.

Following DYNOTEARS, we also illustrate two diagnostic methods for selecting $p$ in a simulated dataset with $p_{\text{true}} = 5$ as shown in Figure 12. In the left panel, the objective function decreases as $p$ increases, plateauing when $p > p_{\text{true}}$, indicating no improvement in model fit with additional complexity. For real-world data, where $p_{\text{true}}$ is unknown, plateaus in the BIC score can guide the selection of $p$. $p$ can also be estimated by analyzing the weights of the inter-slice matrix, as shown in the right panel. When $p$ is unknown, we increment $p$ until the entries of $\boldsymbol{A}_p$ become negligible.

### C.6 PARAMETER SELECTION.

The hyperparameters in DyCAST include the $\ell_1$ sparsity terms, $\lambda_1$ and $\lambda_2$, the scale coefficient of the $\gamma$ stable matrix, and the embedding dimension $r$ of the hidden state. For the sparsity terms $\lambda_1$

and $\lambda_2$, we adopt the values reported in the DYNOTEARS (Pamfil et al., 2020) to ensure a fair comparison. For the remaining parameters, $r$ and $\gamma$, which are specific to DyCAST, we conducted experiments on the simulated dataset, using the F1 score as the evaluation metric.

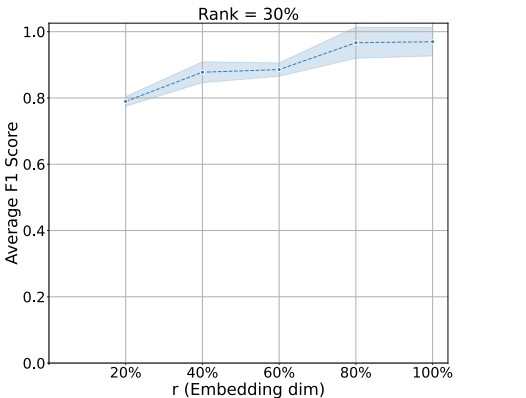
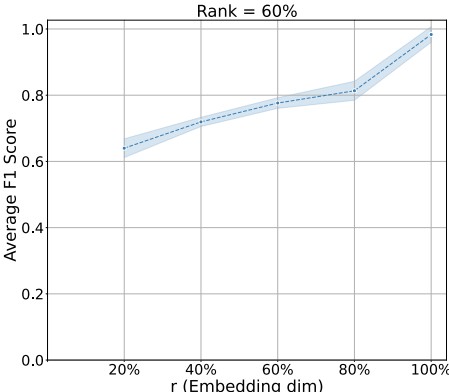

Figure 12: Left panel: F1 score values under different embedding dimensions under dynamic synthetic dataset. Right panel: F1 score values under different stable matrix scale coefficient under dynamic synthetic dataset.

Fang et al. (2024) observed that causal DAGs often exhibit a central structure, resulting in low rank. To evaluate this, we conducted experiments on low-rank and high-rank simulated datasets with $d = 10$, where the rank of the $\boldsymbol{W}_t$ matrix corresponds to 30% and 60% of the number of variables, respectively. The results are presented in Figure 12.

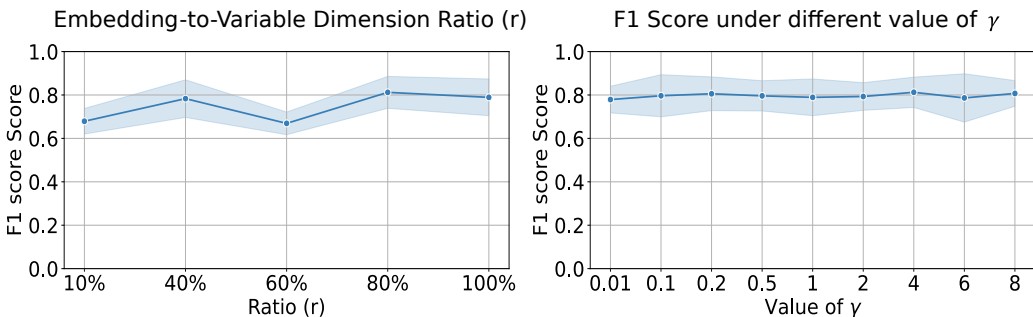

Figure 13: Left panel: F1 score values under different embedding dimensions under dynamic synthetic dataset. Right panel: F1 score values under different stable matrix scale coefficient under dynamic synthetic dataset.

In both low-rank and high-rank scenarios, increasing the embedding dimension $r$ improves DAG accuracy. However, in the low-rank case, even when $r$ is only 20% of the number of variables, the average F1 score still reaches around 0.8. Notably, when the rank is only 10% of the number of variables, as shown in the left panel of Figure 13, DyCAST is relatively insensitive to both hyperparameters. To ensure that the hidden states remain in a low-dimensional space, we select $r$ as 80% of the number of variables. For the scale coefficient $\gamma$, as shown in the left panel of Figure 13, we simply choose a value of 1, as it has minimal impact on the final recovery performance of the intra-slice structure.

## C.7 RUNNING TIMES

In this section, we provide some illustrative running times for different numbers of variables $d$ in Figure 14.

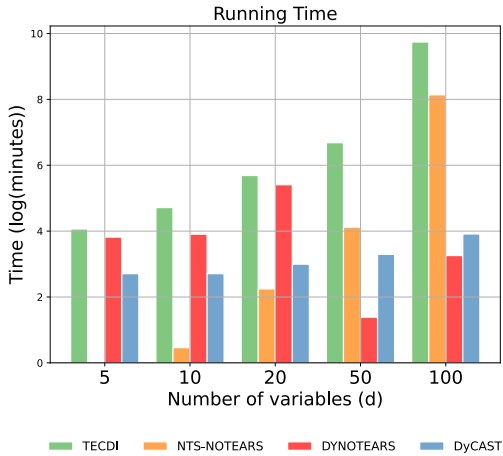

Figure 14: Running times for the simulations from Figure 4 ($N = 500$ samples, $T = 8$ time steps, and $p = 1$ autoregressive term).

We ensure consistent density and edge weights for both intra-slice and inter-slice connections across graphs with varying numbers of variables. The runtime of DyCAST scales gradually with the number of nodes, maintaining a lead of approximately 20%. Notably, TECDI's runtime exceeds 12 hours when the number of variables reaches 100. While DYNOTEARS achieves faster runtimes with fewer variables, it fails to converge when the variable count exceeds 50, leading to shorter but incomplete runs. Additionally, we test the runtime of the DyCAST extension against DyCAST, DyCAST (Not DAG), and DyCAST-CUTS+ on a synthetic dataset. Each model runs for 1000 epochs, and the total runtime is presented in Table 6.

Table 6: Runtime comparison of DyCAST variants on synthetic data with $d = 20$, $T = 8$ time steps.

| Methods | DyCAST | DyCAST (Not DAG) | DyCAST-CUTS+ |
|---|---|---|---|
| Running times | 16.67 min | 16.53min | 17.34min |

The integration with CUTS+ introduces minimal overhead, ensuring DyCAST-CUTS+ remains computationally efficient despite the added complexity of modeling non-linear inter-slice relationships.

## C.8    VISUALIZATION OF THE HIDDEN STATES

We visualize the hidden states of intra-slice structures under a predicted dynamic synthetic dataset and a static synthetic dataset expansion in Figure 15. The left panel shows the hidden states weights of a periodic dynamic intra-slice structure, and the right is the hidden states weights of a static intra-slice structure. We can see that the hidden states under a periodic dynamic intra-slice structure can capture the obvious mutation point, which is the midpoint of the periodicity. Correspondingly, under the static ground truth, the hidden states show the same values.

## C.9    CHALLENGES OF EVALUATING SEPARATE LEARNING ON DYNAMIC SYNTHETIC DATA.

We observe from the experimental results in Figure 4 that NOTEARS and its variants perform poorly on datasets with dynamic modes. To address this, we applied a multi-stage trick to these methods

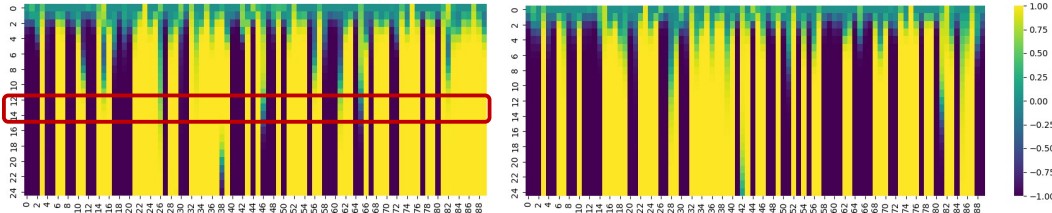

Figure 15: Visualization of the hidden states on synthetic datasets. Left: the hidden states weights of a periodic dynamic intra-slice structure with $d = 10$, $r = 9$, $T = 24$. Right:the hidden states weights of a static intra-slice structure. The vertical axis represents the time step, and the x-axis represents the dimension of the hidden states.

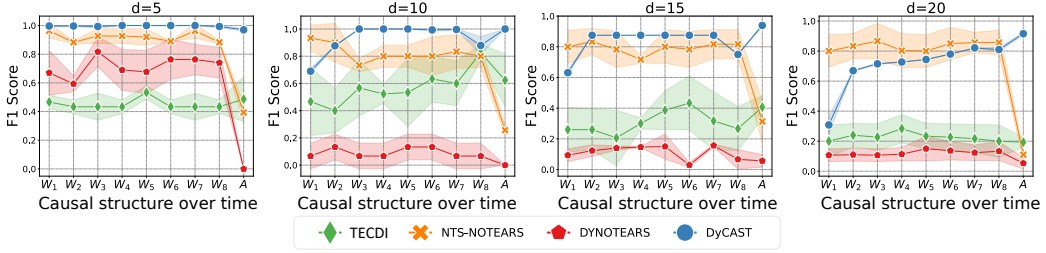

Figure 16: F1 scores for different temporal causal discovery algorithms and different numbers of variables $d \in \{5, 10, 15, 20\}$ on synthetic dataset with dynamic causal structure. Each panel contains results for both intra- and inter-slice structures. Every marker corresponds to the mean performance across 4 algorithm runs, each on a different simulated dataset.

and then compared them with DyCAST. Specifically, we apply the baseline method $T$ times on a dataset of length $T$ to obtain $T$ intra-slice DAGs. Then we compare it with DyCAST obtained from a single run, and the results are shown in Figure 16. We show that the separate learning trick improves the causal discovery performance of DYNOTEARS, NTS-NOTEARS, and TECDI on dynamic datasets but requires running their respective algorithms $T$ times per dataset. Despite this, DYNOTEARS and TECDI with the trick still fall short of or only close to DyCAST. Notably, while NTS-NOTEARS slightly outperforms DyCAST on intra-slice relationships when the variable count is 20, its inter-slice performance remains significantly inferior. In conclusion, DyCAST's ability to jointly learn from all time steps while capturing the dynamic evolution of intra-slice matrices over time provides significantly better results in terms of both accuracy and generalization.

## C.10 NETSIM EVALUATION UNDER AUROC

Due to space constraints in the main text, we report the AUROC results for each method on the NetSim dataset in Table 7. It can be seen that under the AUROC evaluation criteria, DyCAST still achieved the best results.

We also provide the results of running cLSTM, LINGAM, and CUTS on the NetSim dataset, as shown in Table 8. In each column of results, the first column is the AUROC value, followed by the AUPRC value.

## D ADDITIONAL ABLATION STUDY

This section provides additional ablation study results of DyCAST on the NetSim and CausalTime dataset, as shown in Table 9 and Table 10.

We conduct the ablation study by evaluating DyCAST without latent states and $S_0$. Moreover, the causal graph of NetSim and CausalTime is not necessarily DAG, so we added the ablation of DAG constraint. In addition, since DyCAST can be combined with other Granger causality-

Table 7: Performance on NetSim Dataset Under AUROC.

| Metric | DATASET | GC | DYNOTEARS | NTS-NOTEARS | PCMCI | NGC | CUTS+ | LCCM | eSRU | TECDI | DyCAST | DyCAST (Not DAG) | DyCAST-CUTS+ |
|---|---|---|---|---|---|---|---|---|---|---|---|---|---|
| AUPRC | Sim1 | $0.67_{\pm0.11}$ | $0.73_{\pm0.08}$ | $0.64_{\pm0.07}$ | $0.64_{\pm0.12}$ | $0.65_{\pm0.12}$ | $0.75_{\pm0.12}$ | $0.75_{\pm0.07}$ | $0.63_{\pm0.13}$ | $0.67_{\pm0.03}$ | $0.79_{\pm0.01}$ | $0.77_{\pm0.05}$ | **$0.81_{\pm0.02}$** |
| | Sim2 | $0.75_{\pm0.11}$ | $0.81_{\pm0.08}$ | $0.50_{\pm0.04}$ | $0.70_{\pm0.12}$ | $0.68_{\pm0.11}$ | $0.73_{\pm0.09}$ | $0.74_{\pm0.04}$ | $0.66_{\pm0.12}$ | $0.79_{\pm0.04}$ | $0.77_{\pm0.01}$ | **$0.95_{\pm0.07}$** | $0.80_{\pm0.10}$ |
| | Sim3 | $0.77_{\pm0.14}$ | $0.85_{\pm0.07}$ | $0.68_{\pm0.12}$ | $0.73_{\pm0.13}$ | $0.72_{\pm0.12}$ | $0.73_{\pm0.02}$ | $0.75_{\pm0.06}$ | $0.69_{\pm0.12}$ | $0.73_{\pm0.04}$ | $0.75_{\pm0.04}$ | **$0.98_{\pm0.07}$** | $0.80_{\pm0.01}$ |
| | Sim8 | $0.63_{\pm0.11}$ | $0.66_{\pm0.10}$ | $0.49_{\pm0.16}$ | $0.61_{\pm0.11}$ | $0.62_{\pm0.12}$ | $0.75_{\pm0.09}$ | $0.75_{\pm0.07}$ | $0.61_{\pm0.13}$ | $0.58_{\pm0.10}$ | $0.74_{\pm0.02}$ | $0.70_{\pm0.02}$ | **$0.75_{\pm0.05}$** |
| | Sim10 | $0.65_{\pm0.15}$ | $0.69_{\pm0.12}$ | $0.58_{\pm0.05}$ | $0.66_{\pm0.15}$ | $0.65_{\pm0.16}$ | $0.66_{\pm0.09}$ | $0.68_{\pm0.01}$ | $0.64_{\pm0.16}$ | $0.71_{\pm0.05}$ | $0.75_{\pm0.16}$ | $0.76_{\pm0.02}$ | **$0.78_{\pm0.07}$** |
| | Sim11 | $0.72_{\pm0.09}$ | $0.77_{\pm0.04}$ | $0.47_{\pm0.06}$ | $0.68_{\pm0.10}$ | $0.67_{\pm0.09}$ | $0.70_{\pm0.02}$ | $0.72_{\pm0.04}$ | $0.65_{\pm0.10}$ | $0.74_{\pm0.02}$ | $0.69_{\pm0.10}$ | **$0.79_{\pm0.05}$** | **$0.79_{\pm0.07}$** |
| | Sim12 | $0.76_{\pm0.12}$ | $0.83_{\pm0.05}$ | $0.47_{\pm0.02}$ | $0.70_{\pm0.13}$ | $0.68_{\pm0.12}$ | $0.73_{\pm0.06}$ | $0.75_{\pm0.07}$ | $0.66_{\pm0.13}$ | $0.79_{\pm0.02}$ | $0.80_{\pm0.02}$ | **$0.81_{\pm0.05}$** | $0.80_{\pm0.05}$ |
| | Sim13 | $0.62_{\pm0.10}$ | $0.66_{\pm0.08}$ | $0.43_{\pm0.08}$ | $0.59_{\pm0.12}$ | $0.59_{\pm0.12}$ | $0.80_{\pm0.10}$ | **$0.81_{\pm0.10}$** | $0.59_{\pm0.12}$ | $0.68_{\pm0.02}$ | $0.76_{\pm0.12}$ | $0.75_{\pm0.05}$ | $0.80_{\pm0.01}$ |
| | Sim14 | $0.69_{\pm0.10}$ | $0.74_{\pm0.08}$ | $0.39_{\pm0.02}$ | $0.64_{\pm0.11}$ | $0.65_{\pm0.13}$ | $0.76_{\pm0.02}$ | $0.78_{\pm0.06}$ | $0.63_{\pm0.14}$ | $0.67_{\pm0.04}$ | $0.76_{\pm0.13}$ | $0.68_{\pm0.12}$ | **$0.78_{\pm0.06}$** |
| | Sim15 | $0.64_{\pm0.12}$ | $0.68_{\pm0.07}$ | $0.46_{\pm0.07}$ | $0.66_{\pm0.12}$ | $0.68_{\pm0.16}$ | $0.82_{\pm0.07}$ | $0.80_{\pm0.10}$ | $0.65_{\pm0.16}$ | $0.72_{\pm0.04}$ | $0.78_{\pm0.13}$ | $0.63_{\pm0.04}$ | **$0.83_{\pm0.05}$** |
| | Sim16 | $0.60_{\pm0.10}$ | $0.64_{\pm0.07}$ | $0.45_{\pm0.15}$ | $0.59_{\pm0.09}$ | $0.59_{\pm0.11}$ | $0.76_{\pm0.07}$ | $0.75_{\pm0.07}$ | $0.58_{\pm0.12}$ | $0.64_{\pm0.05}$ | **$0.77_{\pm0.07}$** | $0.65_{\pm0.01}$ | **$0.77_{\pm0.06}$** |
| | Sim17 | $0.78_{\pm0.14}$ | $0.87_{\pm0.05}$ | $0.52_{\pm0.02}$ | $0.76_{\pm0.13}$ | $0.77_{\pm0.13}$ | $0.87_{\pm0.10}$ | $0.90_{\pm0.12}$ | $0.73_{\pm0.14}$ | $0.86_{\pm0.02}$ | $0.72_{\pm0.13}$ | **$0.92_{\pm0.13}$** | $0.80_{\pm0.07}$ |
| | Sim18 | $0.68_{\pm0.15}$ | $0.74_{\pm0.08}$ | $0.45_{\pm0.12}$ | $0.64_{\pm0.14}$ | $0.65_{\pm0.16}$ | $0.75_{\pm0.09}$ | $0.74_{\pm0.06}$ | $0.63_{\pm0.16}$ | $0.58_{\pm0.16}$ | **$0.86_{\pm0.16}$** | $0.65_{\pm0.05}$ | **$0.86_{\pm0.05}$** |
| | Sim21 | $0.68_{\pm0.12}$ | $0.74_{\pm0.08}$ | $0.44_{\pm0.10}$ | $0.63_{\pm0.12}$ | $0.64_{\pm0.13}$ | $0.75_{\pm0.11}$ | $0.75_{\pm0.07}$ | $0.62_{\pm0.13}$ | $0.68_{\pm0.03}$ | **$0.81_{\pm0.13}$** | $0.77_{\pm0.02}$ | $0.80_{\pm0.10}$ |
| | Sim22 | $0.62_{\pm0.13}$ | $0.66_{\pm0.07}$ | $0.48_{\pm0.07}$ | $0.61_{\pm0.12}$ | $0.58_{\pm0.13}$ | $0.83_{\pm0.11}$ | **$0.85_{\pm0.05}$** | $0.56_{\pm0.13}$ | $0.83_{\pm0.03}$ | $0.75_{\pm0.13}$ | $0.78_{\pm0.13}$ | $0.78_{\pm0.03}$ |
| | Sim23 | $0.62_{\pm0.09}$ | $0.64_{\pm0.06}$ | $0.43_{\pm0.07}$ | $0.65_{\pm0.11}$ | $0.67_{\pm0.15}$ | $0.65_{\pm0.04}$ | $0.62_{\pm0.01}$ | $0.63_{\pm0.16}$ | **$0.79_{\pm0.02}$** | $0.78_{\pm0.13}$ | $0.70_{\pm0.05}$ | $0.69_{\pm0.01}$ |
| | Sim24 | $0.54_{\pm0.11}$ | $0.53_{\pm0.11}$ | $0.47_{\pm0.12}$ | $0.57_{\pm0.12}$ | $0.55_{\pm0.13}$ | $0.46_{\pm0.02}$ | $0.71_{\pm0.04}$ | $0.55_{\pm0.13}$ | **$0.79_{\pm0.02}$** | $0.70_{\pm0.13}$ | $0.67_{\pm0.05}$ | $0.48_{\pm0.03}$ |
| | Average | $0.67_{\pm0.11}$ | $0.72_{\pm0.08}$ | $0.46_{\pm0.07}$ | $0.65_{\pm0.12}$ | $0.65_{\pm0.13}$ | $0.74_{\pm0.07}$ | $0.76_{\pm0.08}$ | $0.63_{\pm0.14}$ | $0.73_{\pm0.08}$ | **$0.85_{\pm0.13}$** | $0.76_{\pm0.07}$ | $0.77_{\pm0.08}$ |

The best results are in **bold** and the second best are underlined.

Table 8: Performance of other algorithms on the NetSim dataset.

| Metric | DATASET | cLSTM | LiNGAM | SRU | CUTS |
|---|---|---|---|---|---|
| AUPRC & AUROC | Sim1 | $0.64_{\pm0.12}$ / $0.41_{\pm0.14}$ | $0.66_{\pm0.13}$ / $0.43_{\pm0.15}$ | $0.62_{\pm0.13}$ / $0.39_{\pm0.14}$ | $0.74_{\pm0.13}$ / $0.77_{\pm0.14}$ |
| | Sim2 | $0.67_{\pm0.11}$ / $0.28_{\pm0.11}$ | $0.69_{\pm0.12}$ / $0.30_{\pm0.11}$ | $0.66_{\pm0.11}$ / $0.26_{\pm0.10}$ | $0.91_{\pm0.06}$ / $0.84_{\pm0.09}$ |
| | Sim3 | $0.70_{\pm0.12}$ / $0.24_{\pm0.12}$ | $0.73_{\pm0.13}$ / $0.27_{\pm0.13}$ | $0.69_{\pm0.12}$ / $0.22_{\pm0.12}$ | $0.96_{\pm0.05}$ / $0.86_{\pm0.06}$ |
| | Sim8 | $0.61_{\pm0.11}$ / $0.39_{\pm0.11}$ | $0.61_{\pm0.11}$ / $0.36_{\pm0.10}$ | $0.61_{\pm0.13}$ / $0.38_{\pm0.13}$ | $0.65_{\pm0.10}$ / $0.70_{\pm0.08}$ |
| | Sim10 | $0.64_{\pm0.16}$ / $0.42_{\pm0.15}$ | $0.66_{\pm0.15}$ / $0.42_{\pm0.16}$ | $0.63_{\pm0.16}$ / $0.42_{\pm0.15}$ | $0.74_{\pm0.02}$ / $0.77_{\pm0.12}$ |
| | Sim11 | $0.66_{\pm0.13}$ / $0.24_{\pm0.08}$ | $0.68_{\pm0.10}$ / $0.25_{\pm0.08}$ | $0.65_{\pm0.10}$ / $0.23_{\pm0.08}$ | $0.77_{\pm0.09}$ / $0.63_{\pm0.10}$ |
| | Sim12 | $0.66_{\pm0.13}$ / $0.27_{\pm0.11}$ | $0.69_{\pm0.13}$ / $0.30_{\pm0.11}$ | $0.65_{\pm0.13}$ / $0.26_{\pm0.11}$ | $0.78_{\pm0.05}$ / $0.68_{\pm0.10}$ |
| | Sim13 | $0.59_{\pm0.12}$ / $0.47_{\pm0.13}$ | $0.59_{\pm0.12}$ / $0.42_{\pm0.13}$ | $0.58_{\pm0.12}$ / $0.43_{\pm0.19}$ | $0.74_{\pm0.03}$ / $0.80_{\pm0.01}$ |
| | Sim14 | $0.64_{\pm0.11}$ / $0.41_{\pm0.13}$ | $0.66_{\pm0.13}$ / $0.42_{\pm0.13}$ | $0.62_{\pm0.15}$ / $0.39_{\pm0.13}$ | $0.66_{\pm0.05}$ / $0.71_{\pm0.04}$ |
| | Sim15 | $0.67_{\pm0.16}$ / $0.45_{\pm0.20}$ | $0.70_{\pm0.16}$ / $0.48_{\pm0.21}$ | $0.64_{\pm0.16}$ / $0.43_{\pm0.19}$ | $0.60_{\pm0.03}$ / $0.72_{\pm0.04}$ |
| | Sim16 | $0.59_{\pm0.11}$ / $0.45_{\pm0.10}$ | $0.60_{\pm0.11}$ / $0.46_{\pm0.10}$ | $0.58_{\pm0.12}$ / $0.45_{\pm0.10}$ | $0.63_{\pm0.07}$ / $0.76_{\pm0.05}$ |
| | Sim17 | $0.75_{\pm0.14}$ / $0.37_{\pm0.19}$ | $0.78_{\pm0.13}$ / $0.42_{\pm0.19}$ | $0.71_{\pm0.15}$ / $0.34_{\pm0.19}$ | $0.89_{\pm0.01}$ / $0.80_{\pm0.01}$ |
| | Sim18 | $0.64_{\pm0.16}$ / $0.41_{\pm0.16}$ | $0.67_{\pm0.15}$ / $0.43_{\pm0.16}$ | $0.62_{\pm0.16}$ / $0.39_{\pm0.15}$ | $0.63_{\pm0.07}$ / $0.69_{\pm0.09}$ |
| | Sim21 | $0.62_{\pm0.13}$ / $0.40_{\pm0.14}$ | $0.65_{\pm0.13}$ / $0.42_{\pm0.15}$ | $0.61_{\pm0.13}$ / $0.38_{\pm0.13}$ | $0.75_{\pm0.08}$ / $0.73_{\pm0.05}$ |
| | Sim22 | $0.57_{\pm0.13}$ / $0.34_{\pm0.09}$ | $0.60_{\pm0.12}$ / $0.37_{\pm0.09}$ | $0.56_{\pm0.13}$ / $0.34_{\pm0.09}$ | $0.77_{\pm0.10}$ / $0.73_{\pm0.05}$ |
| | Sim23 | $0.67_{\pm0.15}$ / $0.43_{\pm0.19}$ | $0.68_{\pm0.15}$ / $0.47_{\pm0.21}$ | $0.61_{\pm0.17}$ / $0.41_{\pm0.19}$ | $0.66_{\pm0.08}$ / $0.71_{\pm0.07}$ |
| | Sim24 | $0.55_{\pm0.13}$ / $0.34_{\pm0.11}$ | $0.57_{\pm0.12}$ / $0.35_{\pm0.11}$ | $0.54_{\pm0.13}$ / $0.33_{\pm0.11}$ | $0.63_{\pm0.07}$ / $0.73_{\pm0.09}$ |
| | Average | $0.64_{\pm0.13}$ / $0.37_{\pm0.13}$ | $0.66_{\pm0.13}$ / $0.39_{\pm0.12}$ | $0.62_{\pm0.14}$ / $0.36_{\pm0.12}$ | $0.77_{\pm0.07}$ / $0.74_{\pm0.08}$ |

Table 9: Ablation study on NetSim datasets.

| Methods | Sim1 | | Sim2 | | Sim3 | |
|---|---|---|---|---|---|---|
| | AUROC | AUPRC | AUROC | AUPRC | AUROC | AUPRC |
| DYNOTEARS | $0.64_{\pm0.12}$ | $0.41_{\pm0.08}$ | $0.81_{\pm0.08}$ | $0.33_{\pm0.12}$ | $0.85_{\pm0.07}$ | $0.32_{\pm0.13}$ |
| w/o $S_0$ | $0.65_{\pm0.03}$ | $0.68_{\pm0.08}$ | $0.63_{\pm0.01}$ | $0.61_{\pm0.05}$ | $0.71_{\pm0.03}$ | $0.78_{\pm0.10}$ |
| w/o Latent State | $0.73_{\pm0.02}$ | $0.79_{\pm0.01}$ | $0.74_{\pm0.02}$ | $0.82_{\pm0.05}$ | $0.69_{\pm0.05}$ | $0.75_{\pm0.06}$ |
| DyCAST (No DAG) | $0.77_{\pm0.02}$ | $0.79_{\pm0.05}$ | **$0.95_{\pm0.03}$** | $0.85_{\pm0.04}$ | **$0.98_{\pm0.04}$** | **$0.87_{\pm0.01}$** |
| DyCAST | $0.79_{\pm0.01}$ | $0.90_{\pm0.13}$ | $0.77_{\pm0.01}$ | **$0.91_{\pm0.11}$** | $0.75_{\pm0.04}$ | $0.84_{\pm0.04}$ |
| DyCAST-CUTS+ | **$0.81_{\pm0.02}$** | **$0.92_{\pm0.03}$** | $0.80_{\pm0.10}$ | $0.89_{\pm0.12}$ | $0.80_{\pm0.01}$ | $0.85_{\pm0.04}$ |

based methods, we also added the results of DyCAST-CUTS+ for comparison. Specifically, we run DYNOTEARS, DyCAST w/o Latent states, DyCAST w/o $S_0$, DyCAST without DAG constraint, and DyCAST on SimNet and CausalTime.

Table 10: Ablation study on CausalTime datasets.

| Methods | AQI | | Traffic | | Medical | |
|---|---|---|---|---|---|---|
| | AUROC | AUPRC | AUROC | AUPRC | AUROC | AUPRC |
| DYNOTEARS | $0.58_{\pm0.02}$ | $0.65_{\pm0.04}$ | $0.60_{\pm0.07}$ | $0.55_{\pm0.12}$ | $0.65_{\pm0.07}$ | $0.40_{\pm0.13}$ |
| w/o $S_0$ | $0.73_{\pm0.02}$ | $0.61_{\pm0.03}$ | $0.65_{\pm0.01}$ | $0.57_{\pm0.03}$ | $0.74_{\pm0.01}$ | $0.63_{\pm0.10}$ |
| w/o Latent State | $0.80_{\pm0.07}$ | $0.65_{\pm0.04}$ | $0.67_{\pm0.02}$ | $0.60_{\pm0.05}$ | $0.77_{\pm0.06}$ | $0.70_{\pm0.02}$ |
| DyCAST (No DAG) | $0.85_{\pm0.03}$ | $0.70_{\pm0.03}$ | $\mathbf{0.68}_{\pm0.05}$ | $0.60_{\pm0.02}$ | $0.74_{\pm0.01}$ | $0.69_{\pm0.03}$ |
| DyCAST | $0.85_{\pm0.02}$ | $0.82_{\pm0.00}$ | $0.63_{\pm0.01}$ | $0.70_{\pm0.01}$ | $0.81_{\pm0.03}$ | $0.74_{\pm0.03}$ |
| DyCAST-CUTS+ | $\mathbf{0.91}_{\pm0.02}$ | $\mathbf{0.93}_{\pm0.00}$ | $0.65_{\pm0.01}$ | $\mathbf{0.73}_{\pm0.01}$ | $\mathbf{0.84}_{\pm0.01}$ | $\mathbf{0.77}_{\pm0.03}$ |

The results from these experiments provide insights into the contribution of latent states and $S_0$ to the overall performance of DyCAST. We observe that latent states are crucial in DyCAST, with $S_0$ being particularly significant. This is because each row of the matrix derived from $S_0$ fully preserves the influence and dependency information of the variables. Embedding $S_0$ enhances the latent states' ability to capture the intra-slice matrix dynamics. Furthermore, on the NetSim and CausalTime datasets, DyCAST achieves comparable results even without enforcing DAG constraints. However, the absence of DAG constraints makes it harder to identify directed causal relationships, leading to higher AUROC but relatively lower AUPRC.

## E  CASE STUDY: HUMAN3.6M

In this section, We apply DyCAST to a human motion dataset of Human3.6M (Ionescu et al., 2014). The dataset contains activities by 11 professional actors in 17 scenarios, as well as provides accurate 3D joint positions. Although the Human3.6M dataset lacks ground truth for the causal matrix between human joints, human kinematics offers a theoretical foundation for qualitative analysis. Combined with our intuitive understanding of human activities, we can effectively evaluate the plausibility of the dynamic causal DAGs inferred by the algorithm.

We select the Smoking scenario in which the amplitude changes before and after the characters complete the more obvious actions and run DyCAST for analysis, as shown in the right panel of Figure 17. It can be easy to see that, during the smoking action, the key joint system comprising the elbow (No.12 joint), shoulder (No.11 joint), and head (No.10 joint) exhibit dynamic behavior distinct from other joints. These dynamic joints are highlighted with larger red dots in the original trajectory shown in the first row of Figure 17. Our experiment focuses on analyzing the causal relationships within this dynamic system.

We apply DyCAST with $p = 1$, $T = 50$, and embedding dim $r$=12, and also run DYNOTEARS and CUTS+ for a more intuitive comparison. The static method CUTS+ captures constant, inappropriate causal links between the hands and head, even after the smoking action ends, as shown in Figure 17 second row. Similarly, DYNOTEARS, despite identifying the lagged influence of the left elbow and its link to the head, misses the dynamic relationship with the shoulder, as shown in Figure 17 third row. In contrast, DyCAST captures the intra-slice relationships among the elbow, shoulder, and head, as well as the inter-slice dynamics of the elbow during the smoking action. When the smoking action ends at the 40th second, the intra-slice relationships disappear, leaving only the elbow's inter-slice dynamics. This provides a realistic and accurate dynamic representation of the smoking action at the joint level. Thus, in this system with evident complex dynamics, DyCAST, specifically designed for discovering dynamic intra-slice relationships, proves particularly suitable and performs exceptionally well.

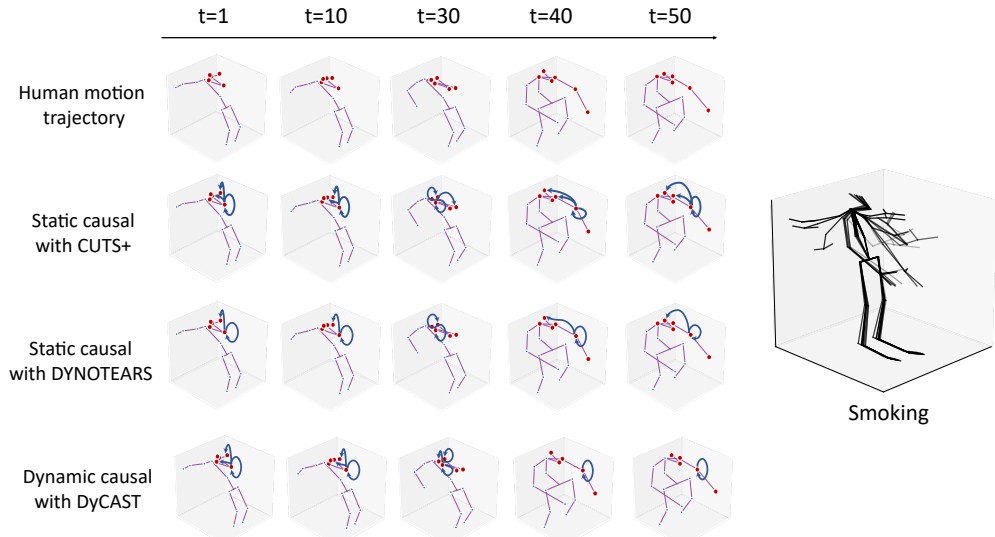

Figure 17: Visualization of the dynamic intra-slice relationships of the Smoking scenario in Human3.6M. Left three rows: Original human motion trajectories across time, alongside static causal relationships of the joint inferred by CUTS+ and DYNOTEARS, and dynamic causal relationships by DyCAST. Right rows: A complete motion trajectory in the Smoking scene across the 50s.

# F DREAM CHALLENGE

**Baseline methods.** For DREAM-3 datasets (Prill et al., 2010), we compare DyCAST with GC (Granger, 1969), DYNOTEARS (Pamfil et al., 2020), NTS-NOTEARS (Sun et al., 2023), PCMCI (Peters et al., 2013), NGC (Tank et al., 2022), CUTS+ (Cheng et al., 2024a), LCCM (Brouwer et al., 2021), eSRU (Khanna & Tan, 2020), TECDI (Li et al., 2023).

In this section, we apply DyCAST to the DREAM-3 network inference challenge, which aims to infer gene regulatory networks from gene expression data. The DREAM-3 dataset consists of 5 independent datasets, each containing 10 time series recordings from 100 genes across 21 time steps. These datasets are widely used for evaluating causal discovery methods. As shown in Table 11, DyCAST-CUTS+ outperforms other methods, demonstrating its effectiveness.

Table 11: Performance on the DREAM-3 dataset.

| Methods | DREAM-3 (d=100) |
|---|---|
| GC | $0.50_{\pm 0.07}$ |
| DYNOTEARS | $0.50_{\pm 0.08}$ |
| NTS-NOTEARS | $0.52_{\pm 0.05}$ |
| PCMCI | $0.55_{\pm 0.03}$ |
| NGC | $0.56_{\pm 0.03}$ |
| CUTS+ | $\underline{0.64}_{\pm 0.07}$ |
| LCCM | $0.50_{\pm 0.03}$ |
| NGM | $0.55_{\pm 0.03}$ |
| eSRU | $0.56_{\pm 0.03}$ |
| CUTS | $0.59_{\pm 0.03}$ |
| TEDCI | $0.50_{\pm 0.08}$ |
| DyCAST | $0.55_{\pm 0.05}$ |
| DyCAST (Not DAG) | $0.57_{\pm 0.05}$ |
| DyCAST-CUTS+ | $\mathbf{0.65}_{\pm 0.05}$ |

From Table 11, we observe that on the real-world dataset with $d = 100$, DyCAST maintains highly competitive performance, significantly outperforming similar methods like DYNOTEARS and NTS-NOTEARS. Given the complexity and nonlinearity of the DREAM-3 dataset, integrating it with the nonlinear CUTS+ further enhances performance, achieving the best results.

## G    IDENTIFIABILITY OF DYCAST

According to the identifiability result [Peters & Bühlmann (2014), Theorem 1], it shows that when the sample size goes to infinity, there is no other DAG structure that can generate the same distribution. When observational data is generated through the structural equation model defined in Eq.(1), the graph structure $\boldsymbol{W}_t$ is identifiable up to a Markov equivalence class and the coefficients $\boldsymbol{W}_t^{i,j}$ can be reconstructed for all $i$ and $j$.

