# OpenReview forum: "DyCAST: Learning Dynamic Causal Structure from Time Series"
_ICLR.cc/2025/Conference — ICLR 2025 Poster_

### Official Review · Reviewer_e4ie · 2024-10-23

**Soundness:** 3
**Presentation:** 2
**Contribution:** 2
**Rating:** 3
**Confidence:** 4

**Summary:**

This manuscript investigates dynamic causal structure in time series using Neural ODEs, a topic of broad interest. The authors reformulate the problem of learning causal structures at each time step as finding the trajectory of a Neural ODE on a directed acyclic graph. Finally, simulations were performed on both synthetic and dynamical models to validate the proposed approach.

**Strengths:**

This paper proposes a method for identifying dynamic causal structure by learning their trajectory using L1 normalization. Both intra-slice and inter-slice structures are learned simultaneously. The method demonstrates higher performance compared to other approaches it is evaluated against.

**Weaknesses:**

The concept of using activity trajectories to identify causal structure seems intriguing. However, due to a lack of clarity, the novelty of the proposed method remains uncertain. The method is not well explained. There are insufficient numerical experiments to support the proposed approach. Several key details are missing. The writing and certain claims need refinement.

**Questions:**

1)	The rationale of Eq.10. What would happen if the concatenation was removed? An ablation study addressing this would be helpful.
2)	Is this approach applicable to large-scale systems? The results presented in the manuscript only cover 5/50/100 nodes.
3)	Line 182, is the correct phrasing "neural underlying ODE" or "underlying Neural ODE"?
4)	No information about the $\lambda$ parameter in Eq. 18.
5)	Why do negative weights appear in Figure 3? This seems unreasonable and counterintuitive. Using a BA synthetic network might be more appropriate.
6)	How does the noise strength affect the performance of the proposed method?
7)	It is unclear why time steps are discussed, but time points are not mentioned. The causal structure detection largely depends on factors like the length of the time series considered, particularly the information at each time point. However, these key details are missing from the manuscript, which raises concerns about the accuracy and applicability of the proposed method.
8)	More rigorous numerical experiments are needed to substantiate the claims made. The current experiments are insufficient to demonstrate the robustness of the proposed method.
9)	I am confused why the authors mentioned time steps, but not mentioned time points at all. Causal structure detection is related to how much information, e.g., the length of time sequences you considered. All these important information is lacking.
10)	I am surprised that the NetSim dataset includes 100 nodes. Can you provide more details about this dataset?
11)	Figure 5 requires sublabels, and in part (b), there seems to be an error (0, 30, 80, 00). Please review and correct it.

---

> ### Author Response · Authors · 2024-11-25
> **Repose, Part I/II**
>
> We thank the reviewer for the valuable comments. Below we address your concerns and questions point by point.
>
> **Weaknesses Part**
>
> 1.*The concept of ... need refinement.*
>
> **Response**: We address your concern as follows:
> - **Clarity and Novelty**: We emphasized the unique integration of latent Neural ODEs and constrained optimization for dynamic causal discovery. All revisions clarifying this are highlighted in blue.
>
> - **Method Explanation**: To enhance understanding, we provided a more detailed description of the methodology, including the key steps and intuition behind the approach, in **Section 3.3, page 6**. Revisions to improve clarity are also marked in blue.
>
> - **Supplement of Experiments**: Additional numerical experiments have been included to validate the proposed approach. These include evaluations on scalability, efficiency, and robustness, as well as experiments with dynamic intra-slice graphs to demonstrate the method's effectiveness. The results are shown in **Appendix C** (pages 15 to 22) and **Appendix E**(page 23).

---

> > ### Author Response · Authors · 2024-11-25
> > **Repose, Part II/II**
> >
> > **Questions Part**
> >
> > 1.*The rationale ... this would be helpful.*
> >
> > **Response**: We agree with your observation regarding the concatenation operation in **Eq. (10)**. If we remove this concatenation, the initial intra-slice matrix $\boldsymbol{W}_0$ would indeed lose important information about the variables affected by each variable when embedding each row. To address this, we have conducted ablation experiments on the synthetic, CausalTime, and NetSim datasets, and the results clearly show that $\boldsymbol{S}_0$ plays a crucial role in capturing the dependencies between variables. These experiments further underscore the importance of this component in the model's performance. The results, are shown in **Table 1 (Section 4.1, page 8)** and **Appendix D(page 23)**.
> >
> > 2.*Is this approach applicable ... 5/50/100 nodes.*
> >
> > **Response**:We would like to kindly highlight the importance of DyCAST's applicability to large-scale datasets, we conducted new experiments with varying numbers of variables $d=\{50,100,200,300\}$, as shown in **Table 5 (Appendix C.1, page 15)**.
> >
> > The results show that baseline algorithms fail to identify causal relationships when the number of nodes reaches 50, while DyCAST maintains an F1 score of approximately 0.4 even when the number of variables increases to 300. This demonstrates DyCAST's ability to effectively handle large-scale systems, highlighting its scalability and robustness in more complex real-world environments.
> >
> > 3.*Line 182, is the ... "underlying Neural ODE"?*
> >
> > **Response**:We fix this as "underlying Neural ODE" in line 182, page 4.
> >
> > 4.*No information ... Eq. 18*
> >
> > **Response**:Specifically,  $\lambda$ is used to control the sparsity of both intra-slice and inter-slice relationships. To ensure a fair comparison, we followed the same setting as DYNOTEARS for $\lambda$ and did not make any special adjustments to it. This choice was made to maintain consistency and ensure comparability with existing methods.
> >
> > 5.*Why do negative weights ... more appropriate.*
> >
> > **Response**:We state clearly that during the data simulation process, we followed the common approach in time series causal discovery models, using the Erdős–Rényi (ER) model to generate the initial intra-slice DAG $\boldsymbol{W}_0$. This model first generates a binary matrix that satisfies the DAG structure. Then, with a certain probability, it randomly assigns values between $[-2.0, -0.5] \cup [0.5, 2.0]$ to the non-zero entries of this matrix, which results in negative weights.
> >
> > This simulation model is intended to better reflect real-world scenarios, where the causal relationship between two variables can be either positive or negative. In fact, the presence of negative weights is a valid case, and the ground truth in the NetSim dataset also includes examples of negative edge weights.
> >
> > 6.*How does the noise ... the proposed method?*
> >
> > **Response**: We conducted extensive experiments under varying levels of noise. Specifically, over datasets with different strengths $T={8, 16, 32, 64, 128}$ and different noise types (Gaussian, Exponential, Gumbel, Uniform), see more details in **Figure 9 (Appendix C.4, page 17)**. The results show that high noise intensity does indeed affect the accuracy of the causal matrix detected by DyCAST, but the impact is most pronounced on the initial intra-slice and inter-slice matrices. This is because there is less data available to learn the initial intra-slice, making it more sensitive to noise.  In contrast, subsequent matrices $\boldsymbol{W}_t$ benefit from more data to learn the dynamic intra-slice relationships, so despite the presence of noise, the impact is relatively small.
> >
> > 7.*It is unclear why time steps ... of the proposed method.*
> >
> > **Response**: We would like to first clarify that "time steps" refer to the length of the time series, in this paper. Following the approach of DYNOTEARS, we generally refer to a specific moment in time as a "time point" when discussing a particular time $t$ in the series.
> >
> > We agree that the length of the time series is crucial for causal structure prediction. To address this, we conducted extensive experiments on dynamic systems with varying time series lengths **(T=8, 16, 32, 64, 128)**. The results show that, with the same sample size, DyCAST achieves higher accuracy for shorter time series. However, the performance slightly decreases for longer time series. Overall, DyCAST is capable of causal discovery for time series with a length $T = 128$ when the sample size is 500. These additional experiments are presented in **Figure 8, Appendix C.4 (page 17)**.

---

> > > ### Author Response · Authors · 2024-11-25
> > > **Repose, Part II/II**
> > >
> > > **Questions Part (continuing)**
> > >
> > > 8.*More rigorous numerical experiments ... of the proposed method.*
> > >
> > > **Response**:  We agree that DyCAST requires more rigorous numerical experiments to substantiate the claims made. Therefore, we have added several additional quantitative experiments. These include experiments with varying numbers of variables (**Appendix C.1**), different sample sizes (**Appendix C.2**), different sequence lengths (**Appendix C.3**), noise strengths and types (**Appendix C.4**), unknown autoregressive order (**Appendix C.5**), varying causal ranks (**Appendix C.6**), and Running time (**Appendix C.7**). To further explain the success of our method, we have included ablation studies (**Appendix D**) on the concat operation, both on synthetic and real-world datasets. Additionally, we validated the necessity of dynamic intra-slice using a new human 3.6M dataset (**Appendix E**).
> > >
> > > 9.*I am confused why the ... information is lacking.*
> > >
> > > **Response**: To address this, we have added experiments with time series lengths of (**T = {8, 16, 32, 64, 128}**) to evaluate DyCAST's performance on datasets of varying lengths. Additionally, we would like to clarify that in this paper, "time steps" refers to the length of the observed time series, i.e., how many snapshots of data have been collected. Following the convention in DYNOTEARS, we use the term "time point" to refer to a specific point in time, denoted as $t$, within a given time series. Our focus is on modeling the dynamics of the intra-slice at different time points/snapshots/time steps.
> > >
> > > 10.*I am surprised that ... details about this dataset?*
> > >
> > > **Response**: We thank the reviewer for the advice. NetSim results were reported for node counts of (d = \{5, 10, 15\}) only. We change this to "d = \{5, 10, 15\}" in line 425, page 8.
> > >
> > > 11.*Figure 5 requires ...review and correct it.*
> > >
> > > **Response**: We have addressed this issue and made the necessary corrections in the revised version.

---

> > > ### Comment · Reviewer_e4ie · 2024-11-26
> > >
> > > Thank you for these reponses. After reviewing the current version, I believe it is not yet suitable for acceptance, as several concerns remain.
> > >
> > > The most significant concern is regarding the practical application. The largest number of variables considered in the experiments is 300, and most experiments involve {5, 10, 15} variables, which I do not consider to be "large-scale" systems.
> > >
> > > There is one ambiguity regarding the time series data. Readers who are not familiar with the specifics might find it difficult to understand the dataset. If $T$ represents the time series length, then it is unclear what the time interval between data points is. More clarity on this manuscript is crucial, as it could affect both the interpretation and scalability of the methodology.

---

> ### Author Response · Authors · 2024-11-26
> **Repose**
>
> **Questions Part**
>
> 1.**On the large-scale systems**. First, we would like to clarify that 300 variables are already considered large-scale in causal discovery tasks. Prominent methods such as DYNOTEARS [1], NTSNOTEARS [2], TECDI [3], CUTS [4], CUTS+ [5], NGM [6], NGC [7], and LCCM [8] typically benchmark their main experiments on datasets with around 20 variables. Our method, however, has been tested with variable counts exceeding 100, 200, and even 300, maintaining an impressive F1 score of 0.56. In contrast, similar approaches often fail to produce meaningful results with as few as 50 variables, underscoring the robustness and scalability of our approach.
>
> Real-world causal discovery datasets with ground truth typically involve fewer than 50 variables. For instance, NetSim[9] ($d=5, 10, 15$), CausalTime [10] ($d=20, 20, 36$), and RiverFlow [11] ($d=3$) are among the standard benchmarks. Of these, CausalTime, the most recent and mainstream dataset published in ICLR2024, exemplifies this trend.
>
> To further validate our method's effectiveness, we conducted additional experiments on DREAM3 [12], a gene expression and regulation dataset with $d=100$, as shown in the following Table:
>
> |Methods|Dream-3(d=100)|
> ---|---
> PCMCI | 0.55±0.03 |
> NGC | 0.56±0.03 |
> eSRU | 0.56±0.03 |
> SCGL | 0.53±0.03 |
> LCCM | 0.50±0.03 |
> NGM | 0.55±0.03 |
> CUTS | 0.59±0.03 |
> CUTS+ | 0.64±0.07 |
> DyCAST | 0.55±0.05 |
> DyCAST(Not DAG) | 0.57±0.05 |
> DyCAST-CUTS+ | 0.65±0.05 |
>
> The results demonstrate that DyCAST achieved an AUROC of 0.55, DyCAST (Not DAG) achieved 0.57, and DyCAST-CUTS+ achieved 0.65, outperforming CUTS+ alone. These findings illustrate the ability of DyCAST to handle complex, nonlinear real-world datasets, achieving competitive performance even at larger scales.
>
> 2.**On the time interval between data points**. Regarding the time interval between data points, we adhere to the same approach as DYNOTEARS [1], NTSNOTEARS [2], TECDI [3], CUTS [4], CUTS+ [5], NGM [6], NGC [7], and LCCM [8]. Specifically, we utilize a coherent time series without interval sampling between data points for synthetic datasets.
>
> For real-world datasets, we also utilize a coherent time series without interval sampling between data points, and we maintain the inherent interval sampling of the respective datasets:
>
> - **NetSim**: No interval sampling is applied between time intervals.
> - **CausalTime**:
>   - In the **AQI subset**, the interval between data points is hourly.
>   - In the **Traffic subset**, the interval is 5 minutes.
>   - In the **Medical subset**, the interval is 2 hours.
> - **Human3.6**: The interval between data points corresponds to one frame.
>
> We follow the established settings outlined in the NTSNOTEARS and CUTS+ papers, ensuring consistent comparisons when running experiments on all algorithms. As we do not incorporate interval sampling into our method, our experiments remain aligned with the natural intervals of the datasets or the standard practices of prior works. We will include the corresponding description in the revision.
>
> ---
> **Reference**
>
> [1] Pamfil et al. (2020) DYNOTEARS: Structure Learning from Time-Series Data.
>
> [2] Sun et al. (2023) NTS-NOTEARS: Learning Nonparametric DBNs With Prior Knowledge.
>
> [3] Li et al. (2023) Causal Discovery in Temporal Domain from Interventional Data.
>
> [4] Cheng et al. (2023) CUTS: Neural Causal Discovery from Irregular Time-Series Data.
>
> [5] Cheng et al. (2024) CUTS+: High-Dimensional Causal Discovery from Irregular Time-Series.
>
> [6] Bellot et al. (2022) Neural graphical modelling in continuous-time: consistency guarantees and algorithms.
>
> [7] Tank et al. (2021) Neural Granger Causality.
>
> [8] Brouwer et al. (2021) Latent convergent cross mapping.
>
> [9] Smith et al. (2011) Network modelling methods for FMRI.
>
> [10] Cheng et al. (2024) CausalTime: Realistically Generated Time-series for Benchmarking of Causal Discovery.
>
> [11] Rohekar et al. (2023) From temporal to contemporaneous iterative causal discovery in the presence of latent confounders.

---

> > ### Comment · Reviewer_e4ie · 2024-11-27
> >
> > Thank you. However, I must express my concerns regarding the accuracy, novelty of the results, and the claims about handling large-scale systems. If $10^2$ variables are considered large-scale, the performance of this method appears to fall short compared to the state-of-the-art. For instance, some references (listed below)
> >
> > [1] Zhang, Wei, et al. "Cause: Learning granger causality from event sequences using attribution methods." International Conference on Machine Learning. PMLR, 2020. [2] Xiao, Shuai, et al. "Learning time series associated event sequences with recurrent point process networks." IEEE transactions on neural networks and learning systems 30.10 (2019): 3124-3136.
> >
> > For example, some references (e.g., Ru, Xiaolei, et al. "Attentive transfer entropy to exploit transient emergence of coupling effect." NeurIPS 2023) have tested methods on systems with $10^4$ variables.
> >
> > In comparison, the method presented in this manuscript does not appear to offer particularly novel insights or achieve state-of-the-art performance.

---

> ### Author Response · Authors · 2024-11-27
> **Response**
>
> We would like to clarify that the key contribution of DyCAST lies in providing a DBN-based full-time causal graph that captures both intra-slice and inter-slice relationships for each time point in a multivariate time series. Importantly, the **dynamic intra-slice graphs** always satisfy the directed acyclic graph (DAG) constraint, distinguishing our work from approaches focused on Granger Causality [1-3]. Granger Causality typically provides a **static summary graph** and does not enforce the DAG constraint.
>
> We acknowledge that the cited work [3] can handle $10^4$ variables; however, its primary focus is on inferring **coupling relationships between observed variables** and reconstructing the coupled network from time series data. In contrast, our approach emphasizes inferring **causal relationships between variables while strictly adhering to DAG constraints**. This fundamental distinction explains why the two methods are designed to handle different scales of variables. Furthermore, the listed literature [1] and [2] primarily addresses causality in **discrete event sequences**, where causal relationships are easier to discover due to explicit event representations and clear timestamps. In contrast, **continuous numerical time series** introduces greater complexity, making causal discovery significantly more challenging. This complexity is especially pronounced when scaling to large systems, where dynamic causal discovery remains a demanding task.
>
> we thank the reviewer for pointing out these papers, and we will add an additional related works section in the revision to discuss the papers you mentioned.
>
> **Reference**
>
> [1]  Zhang, Wei, et al. (2020) "Cause: Learning granger causality from event sequences using attribution methods." ICML.
>
> [2] Xiao, Shuai, et al. (2019) "Learning time series associated event sequences with recurrent point process networks. TNNLS.
>
> [3] Ru, Xiaolei, et al. (2023) "Attentive transfer entropy to exploit transient emergence of coupling effect." NeurIPS.

---

> ### Comment · Reviewer_e4ie · 2024-11-27
>
> Quick comments:
>
> 1. The condition of a directed acyclic graph (DAG) should be much easier to deal with. Why, then, does even a basic, more general causality method like Granger causality fail in your setting? And also attentive transfer entropy?
> 2. If I was wrong, please clarify: I do not fully understand the phrase "continuous numerical time series." Authors have specified the time length and time interval, which suggests discrete time points rather than truly continuous time.
>
> Importantly, I am strongly concerned about the accuracy and seriousness of this manuscript. The authors have surprisingly miswritten reference details, including the year and journal name, even after I pointed this out in earlier comments ALREADY!
>
> So, I cannot trust any results of this manuscript.

---

> ### Author Response · Authors · 2024-11-27
> **Response, Part I/II**
>
> 1.*The condition of a directed acyclic graph (DAG) ... attentive transfer entropy?*
>
> **Response**: We sincerely apologize for our lack of rigor in checking reference details. It was indeed due to our negligence that the journal number of reference [1] and the year number of reference [2] were incorrectly marked. We have modified the corresponding answers above.
>
> Granger causality-based models typically assume that each sampled variable $x_{i,t}$ is generated by a Structural Causal Model (SCM) with additive noise:
>
> $$X_{i,t} = f_i(X_{1, t-p:t-1}, X_{1, t-p:t-1}, .... ,X_{d, t-p:t-1})$$
>
> where any variable in \(\{\boldsymbol{X}_{t-p}, \ldots, \boldsymbol{X}_{t-1}\}\) may effectively influence \(\boldsymbol{X}_t\). Under this assumption, no additional constraints are imposed on the summary causal graph. To account for sparsity, regularization terms (e.g., $L_1$ or $L_2$) are applied.
>
> Furthermore, Granger causality-based methods do not model the intra-slice causal term $\boldsymbol{X}_t \boldsymbol{W}_t$, thereby completely avoiding the DAG constraint. This omission ignores causal relationships within slices, limiting their ability to capture the full dynamics of $\boldsymbol{W}_t$. At the same time, these methods provide only a summary causal graph, identifying whether a causal relationship exists between $\boldsymbol{X}_i$ and $\boldsymbol{X}_j$, but not specifying if $\boldsymbol{X}_{i,t}$ is causally influenced by a lagged value $\boldsymbol{X}_{j,t-k}$. Consequently, even advanced approaches like Attentive transfer entropy cannot recover the full-time causal graphs.
>
> In contrast, our model further incorporates the data $\boldsymbol{X}_t$ at time $t$ and assumes that when $(\boldsymbol{W}_t$ satisfies the DAG constraint, thus the model becomes:
>
> $$X_{i,t} = X_t W_t + X_{t-1} A_{1} + ... + X_{t-p}A_{p} $$
>
> Here, the intra-slice matrix $\boldsymbol{W}_t$ must satisfy the DAG constraint at all times. While full-time DAG causal graphs are generally sparser than directed graphs (as they exclude bidirectional edges and cycles), enforcing the DAG constraint is computationally challenging. The widely-used acyclicity constraint was introduced by Zheng et al [3]. defines $h(\boldsymbol{W}) = \text{Tr}(e^{\boldsymbol{W} \odot \boldsymbol{W}}) - d$, where $\text{Tr}$ denotes the matrix trace . In practice, enforcing this constraint requires careful tuning of the augmented Lagrangian parameters [4]. Moreover, as the penalty coefficient approaches infinity to ensure acyclicity, numerical instabilities, and ill-conditioning issues arise, as shown empirically by Ng et al. [5].
>
> Our model further requires that $\boldsymbol{W}_t$ dynamically satisfy the DAG constraint at every time step, making optimization significantly more challenging. This is a key reason our method cannot scale to $10^4$ variables.

---

> ### Author Response · Authors · 2024-11-27
> **Response, Part II/II**
>
> 2.*If ... truly continuous time.*
>
> **Response**:  In observed event sequences, each node represents a discrete event type (e.g., A, B, C). For instance, the ATMs dataset (which is used in reference [6]) contains event logs from 1,554 ATMs, including failure tickets (TIKTs) and error reports. These logs capture discrete states such as device identity, timestamps, message content, priorities, codes, and actions. This discrete nature focuses on direct connections between events, facilitating causal reasoning.
>
> In contrast, the time series data we handle consists solely of continuous real values. For example, the CausalTime dataset records traffic flows in real values at 20 intersections, and the Human3.6 dataset provides three-dimensional spatial coordinates for human joints. This type of data is inherently noisy, continuously variable, and influenced by complex periodicities, trends, and multivariate dependencies, making it harder to define causal relationships. These complexities often obscure the underlying causality, significantly increasing the difficulty of causal discovery.
>
> ---
> **Reference**
>
> [1]  Zhang, Wei, et al. (2020) "Cause: Learning granger causality from event sequences using attribution methods." International Conference on Machine Learning.
>
> [2] Ru, Xiaolei, et al. (2023) "Attentive transfer entropy to exploit transient emergence of coupling effect." Advances in Neural Information Processing Systems.
>
> [3] Zheng, Xun, et al.  (2018) "Dags with no tears: Continuous optimization for structure learning." Advances in neural information processing systems.
>
> [4] Ng, Ignavier, et al. (2020) "On the role of sparsity and dag constraints for learning linear dags." Advances in Neural Information Processing Systems.
>
> [5] Ng, Ignavier, et al. (2022) "On the convergence of continuous constrained optimization for structure learning." International Conference on Artificial Intelligence and Statistics.
>
> [6] Xiao, Shuai, et al. (2019) "Learning time series associated event sequences with recurrent point process networks. IEEE transactions on neural networks and learning systems.

---

### Official Review · Reviewer_gcbX · 2024-10-29

**Soundness:** 3
**Presentation:** 3
**Contribution:** 2
**Rating:** 6
**Confidence:** 3

**Summary:**

This work presents DyCAST, a novel framework leveraging Neural Ordinary Differential Equations (Neural ODEs) to learn dynamic causal structures, showing superior performance compared to existing models across various datasets.

**Strengths:**

1. The algorithm effectively supports dynamic contemporaneous structures.

2. The experimental results are compelling, though there are inconsistencies in the choice of baseline methods across different sections (see weaknesses).

**Weaknesses:**

1. While DyCAST claims to support dynamic intra-slice graphs, the ground truth causal graphs in the Netsim and CausalTime datasets do not contain such dynamics. Therefore, although I appreciate the results in Figure 6, I am concerned that the claim is not fully validated in the quantitative experiments.

2. Several baseline methods are included in the CausalTime experiments but not in the Netsim experiments. Additionally, the baseline choices differ between the Netsim and Synthetic experiments. Is there a specific reason for this? If not, more baseline comparisons should be included. If time is limited during the rebuttal period, incorporating the most recent works would be acceptable.

**Questions:**

1. As someone who is not an expert in Neural ODEs, I find the model presented in Equation (4) somewhat unconvincing. Why not learn $W_t$ based on the data at each time step? Are there theoretical justifications or ablation studies to support this modeling choice? If this matter is cleared up, I will be happy to raise my score.

2. Are there additional examples that demonstrate the necessity of dynamic intra-slice graphs?

---

> ### Author Response · Authors · 2024-11-25
> **Repose, Part I/II**
>
> We thank the reviewer so much for the insightful comments and appreciation of our work. We address your concerns and questions in the following point by point.
>
> **Weaknesses Part**
>
> 1.*While DyCAST ... experiments.*
>
> **Response**: Actually, in the original version of the manuscript, we have already introduced a synthetic dataset with dynamic intra-slice graphs to evaluate our model quantitatively, as described in  (**Figure 3**) and **Section 4.1 (pages 7–8)**. This dataset features 7 dynamic intra-slice graphs, 5 variables, and an autoregressive order of 1. DyCAST achieves results closely matching the ground truth. Additionally, extensive experiments on datasets with varying numbers of variables, compared to DYNOTEARS and its variants (**Figure 4**), demonstrate DyCAST's effectiveness in capturing dynamic intra-slice graphs.
>
> Given the limited availability of real-world dynamic causal graph datasets, we run DyCAST on a subset of scenes from the Human3.6M dataset and analyze the resulting dynamic intra-slice data, as shown in **Figure 17 (Appendix E, page 23)**. Our experiments demonstrated that DyCAST was able to effectively capture the causal relationships between body joints as they change over time. During the smoking motion, CUTS+ captures the causal link between the hands and head but keeps it static, even after smoking ends. DYNOTEARS identifies the lagged influence of the left elbow and its link to the head but misses the dynamic relationship with the shoulder. In contrast, DyCAST dynamically captures and adjusts the causal relationships, reflecting the natural dynamics of smoking, and demonstrating its superiority in real-world scenarios.
>
> 2.*Several baseline methods ... recent works would be acceptable.*
>
> **Response**:We agree that clarifying the choice of baselines is essential for a thorough evaluation of DyCAST.
>
> In our synthetic experiments, the ground truth provided includes both intra-slice matrices at each time step and static inter-slice matrices. Since only DYNOTEARS and its variants are capable of outputting this type of ground truth, we focused our comparisons on these methods, namely DYNOTEARS, NTS-NOTEARS, and TECDI. This is why we limited our baseline comparisons to these methods in the synthetic dataset experiments.
>
> For the CausalTime and NetSim datasets, the ground truth is in the form of summary graphs, which makes it possible to compare against a wider range of causal discovery methods. In the revised manuscript, we have aligned the baseline comparisons for these datasets by adding results from other top-performing methods. Specifically:
>
> - For the NetSim dataset, we have added the results for **GC, CUTS+, LCCM, DyCAST (Not DAG) and DyCAST-CUTS+**. We report the results in **Table 2, Section 4.2 (pages 9)** and **Table 6, Appendix C.10 (pages 21)**.
>
> - For the CausalTime dataset, we have included the results for TECDI. We report the results in **Table 3, Section 4.3 (pages 10)**.
>
> - These updates ensure that we now provide a more comprehensive comparison of DyCAST's performance on both synthetic and real-world datasets. We appreciate your suggestion, as it has helped us improve the clarity and depth of our baseline analysis.

---

> > ### Author Response · Authors · 2024-11-25
> > **Repose, Part II/II**
> >
> > **Questions Part**
> >
> > 1.*As someone who is ... I will be happy to raise my score.*
> >
> > **Response**: We would like to kindly highlight that learning the matrix within each slice separately can reduce the number of dynamic interactions that need to be modeled, but **increases the model computation and running time by T times** (T is the number of time steps).
> >
> > In the original version of the manuscript, we conducted an experiment comparing DyCAST with DYNOTEARS, NTSNOTEARS, and TECDI with **separate learning trick**, as described in **Figure 16 (Appendix C.9, page 16)**. We show that the separate learning trick improves the causal discovery performance of DYNOTEARS, NTSNOTEARS, and TECDI on dynamic datasets but requires running their respective algorithms $T$ times per dataset. Despite this, DYNOTEARS and TECDI with the trick **still fall short** of or only close to DyCAST. Notably, while NTSNOTEARS slightly outperforms DyCAST on intra-slice relationships when the variable count is 20, its inter-slice performance remains significantly inferior. In conclusion, although the separate learning tricks have their advantages in terms of simplicity, DyCAST's ability to jointly learn from all time steps while capturing the dynamic evolution of intra-slice matrices over time provides significantly better results in terms of both accuracy and generalization.
> >
> > 2.*Are there additional ... dynamic intra-slice graphs?*
> >
> > **Response**:To further demonstrate the importance of modeling dynamic intra-slice relationships, we conducted additional experiments on the Human3.6M dataset, which includes 17 different scenes with 11 people and 16 joints, capturing the coordinates of human body joints over time, as shown in **Figure 17 Appendix E, (page 23)**.
> >
> > In these experiments, we applied DyCAST and analyzed the dynamic intra-slice matrices obtained from the model. The results revealed that DyCAST successfully captures the reasonable time-varying causal relationships between the joints, reflecting the dynamic nature of the human body’s movement. In contrast, the static method CUTS+ captures the inappropriate constant causal links between the hands and head even after the Smoking action ends. On the other hand, the static method DYNOTEARS identifies the lagged influence of the left elbow and its link to the head but misses the dynamic relationship with the shoulder. In contrast, DyCAST dynamically captures and adjusts the causal relationships, reflecting the natural dynamics of smoking, and demonstrating its superiority in real-world scenarios.
> >
> > This result further validates the effectiveness of capturing dynamic intra-slice relationships, as it allows us to model how the causal dependencies between joints change over time, providing deeper insights into the temporal dynamics of the system.

---

> > > ### Comment · Reviewer_gcbX · 2024-11-26
> > >
> > > I thank the authors for the response. I am in general satisfied with the update. I have raised my score.

---

> > > > ### Author Response · Authors · 2024-11-26
> > > >
> > > > We thank the reviewer so much for the quick reply and we really appreciate the reviewer for raising score.

---

### Official Review · Reviewer_AYt4 · 2024-10-30

**Soundness:** 2
**Presentation:** 3
**Contribution:** 3
**Rating:** 8
**Confidence:** 4

**Summary:**

This paper introduces DyCAST, a method for determining time-varying causal graphs using constrained neural ODEs. In particular, DyCAST determines the causal graph for a set of variables $\mathbf{X}_t$ as a linear combination of dynamic "intra-slice" causes, $\mathbf{X}_t \mathbf{W}_t$, and static "inter-slice" causes, $\sum\_{k=1}^P \mathbf{X}\_{t-k} \mathbf{A}\_k$, plus additive noise. The dynamics of $W_t$ are modeled as a constrained neural ODE, which is extended to a latent ODE using the encoder-process-decoder framework. The latent state models an autonomous system, but may be concatenated with the time $t$ in decoding for systems with periodic components. Finally, a sparsity penalty is added, and DyCAST is combined with CUTS(+), a Granger-causal method for time-varying inter-slice graphs. Extensive experiments are performed on some limited synthetic data and several real or simulated data sets, where DyCAST with CUTS(+) is typically the best-performing method in terms of AUROC and AUPRC.

**Strengths:**

- The issue of time-varying causal structure is under-addressed and practical problem. Using constrained neural ODEs is an elegant solution, and the results are impressive.
- The paper is well-written and introduces concepts from neural ODEs very well. The included illustrations are helpful, and the figures are instructive.
- Incorporating time-varying structures can be extremely interpretable in some scenarios. For example, I found Figure 6 very compelling, with causal influences smoothly following the time of day.

**Weaknesses:**

**On the Combination With Other Methods**

The combination of DyCAST with CUTS+ is one of the more important contributions in the experiments section of the paper, but is not discussed much. This is especially difficult to me because CUTS+ is a Granger-causal method, and to the best of my knowledge does not actually learn what are referred to as "inter-slice" graphs, except in the trivial $p=1$ case.

It would improve the presentation of the paper to include more details of how DyCAST can be combined with other methods like CUTS+, and also how this changes the computational requirements and potential interpretations. Moreover, as an aside, the paper inconsistently refers to DyCAST-CUTS as using CUTS or CUTS+.

**A Lack of Ablation on Most Datasets**

As mentioned in the strengths, the performance of DyCAST is impressive, and it is apparent that it is beneficial in dynamic problems. That said, I find the ablation study in Section 4.1 very interesting, and would like to see a similar experiment in the NetSim and CausalTime experiments. In particular, from Table 1, it is clear that the latent state provides at least some benefit, even in static problems, but this isn't explored outside of the simplistic synthetic case. I view this as critical to understanding *why* DyCAST is successful.

**Minor Typos**

There are several typos throughout the manuscript; ones I noticed include:
- (Page 2, Line 085) NOTEATS should be NOTEARS.
- (Page 2, Line 087) DYNOTEATS should be DYNOTEARS.
- (Page 3, Line 124) I think $X_{t-k}^j$ should be bolded.
- (Equation 18) In the first summand, there is an $\rvert$ instead of $\rVert$.
- (Page 8, Line 398) NTS-NOTEATS should be NTS-NOTEARS.
- (Page 8, Line 398) TEDCI should be TECDI.
- (Page 10, Line 404) "In future" should be "In the future".
- (Page 13, Line 657) Erdős–R'enyi should be Erdős–Rényi.

**A Few Other Minor Points**

- (Related Work) The approach of CCM is of course much different than the proposed approach -- nevertheless, latent CCM (Brouwer et al., 2021) seems like a relevant application of neural differential equations to causal learning to be mentioned in related work.
- (Implementation and Code) While I think the details in Appendix A would be sufficient to reproduce most of the paper's experiments, I hope that the authors consider releasing their code with a finalized version of the paper, as I think it strengthens the impact and reproducibility of the paper.

**Questions:**

- My main question, outlined in weaknesses, is **to what extent can the performance of DyCAST be attributed to the latent state**? For example, in Table 1, the latent state is shown to significantly improve performance; could similar ablation studies be performed in the other experiments?
- There is a brief mention of running time on page 7 ("moreover, DyCAST reduces the running time by more than 20% across various variable dimensions"). This claim is not particularly clear to me: **with respect to what is the running time reduced? At what variable dimension is this noticeable?** Similar claims are made in Appendix A.2.1, but without any specific figures. In general, I think it would improve the manuscript to report the run time of DyCAST with respect to the baselines.
- **What are implementation details for DyCAST-CUTS?** And how does this affect runtime? How are dynamic inter-slice graphs handled outside of the $p=1$ case?
- It seems like the most straightforward approach in the current framework for inter-slice dynamics would be another (independent?) neural ODE on $\mathbf{A}$. Is there a reason why this doesn't work?
- It seems surprising to me that the $F_1$ score is so insensitive to $r$ -- **how would the authors interpret this? Would they expect very small $r$ to be more detrimental in problems with faster dynamics?**
- The methods for comparison are different in NetSim and CausalTime datasets; for example, TECDI is included in NetSim but not CausalTime, and vice versa for LCCM. **Is there a reason for the differing benchmark methods in these sections?** Since both of these methods appear to be the closest competitors to DyCAST, it would be great to standardize the benchmarks, and include both TECDI and LCCM in all comparisons.

---

> ### Author Response · Authors · 2024-11-25
> **Repose, Part I/II**
>
> We thank the reviewer for the valuable comments. Below we address your concerns and questions point by point. We also make the corresponding changes in the revision following your suggestions.
>
> **Weaknesses Part**
>
> 1.*The combination of DyCAST ... using CUTS or CUTS+.*
>
>
> **On the combination with other methods:** We sincerely thank the reviewer for recognizing the flexibility of DyCAST. In its foundational model (**Eq. (3)**), DyCAST assumes that inter-slice dependencies are **linear and fixed**. While this is a reasonable simplification in many cases, real-world data often exhibit complex **non-linear relationships**.
>
> To handle this, we extend Eq. (3) by replacing the linear inter-slice term with a more **flexible function $\varphi$**, allowing for non-linear modeling:
> $$
> \boldsymbol{X}_t = \boldsymbol{X}_t\boldsymbol{W}_t+\varphi(\boldsymbol{X}_{t-p:t-1})+\boldsymbol{Z}_t $$
>
> where $\varphi$ parameterizes the **non-linear inter-slice causal relationships**.This framework can accommodate a wide range of models specifically designed to capture such relationships.
>
> Therefore, with such an extension of DyCAST, we can easily integrate it into CUTS+, a method that focuses on learning inter-slice causal relationships, characterization of $\varphi (\boldsymbol{X}_{t-p:t-1})$ in **Eq(20) (Section 3.3, page 6)**. While CUTS+ is a Granger causality-based model and produces an inter-slice **summary graph**, it is capable of approximating a **full-time graph** when the lag order **p=1**, which aligns with our requirements (this implies that when the autoregressive order **p>1**, we can only derive the summary graph for the lagged relationships of $\boldsymbol{X}_t$ with respect to $\boldsymbol{X}_t$).
>
> A detailed explanation of this modification and its integration into DyCAST is now provided in **Section 3.3, Extension of DyCAST, page 6**.
>
> - The mention of "CUTS" instead of "CUTS+" in the main text and tables was an error. We confirm that all experiments were conducted using **CUTS+**, which is both more effective and efficient than its predecessor. We have thoroughly reviewed the manuscript and corrected this inconsistency throughout the revised submission.

---

> > ### Author Response · Authors · 2024-11-25
> > **Repose, Part I/II**
> >
> > **Weaknesses Part  (continuing)**
> >
> > 2.*As mentioned in the strengths ... is successful.*
> >
> > **On the lack of ablation**: We supplement ablation studies on real-world data to provide valuable insights into the contributions of individual components to the model's performance. These additional experiments are presented in **Appendix D, page 23**. We show that:
> > - **Ablation study on NetSim dataset:**
> >    - We evaluated DyCAST and its ablated versions on subsets of **NetSim** with  **d = 5, 10, 15** variables, which only has these three variable types. The results are summarized in **Table 8** (**Appendix D, page 23**).
> >
> > Methods|Sim1| |Sim2| | Sim3||
> > ---|---|---|---|---|---|---
> >  || AUROC | AUPRC | AUROC | AUPRC| AUROC | AUPRC
> >  DYNOTEARS | 0.64±0.12| 0.41±0.08|0.81±0.08|0.33±0.12| 0.85±0.07| 0.32±0.13
> >  w/o $\boldsymbol{S}_0$|0.65±0.03| 0.68±0.08| 0.63±0.01| 0.61±0.05|0.71±0.03| 0.78±0.10
> >  w/o Latent State | 0.73±0.02 | 0.79±0.01 |0.74±0.02 | 0.82±0.05| 0.69±0.05| 0.75±0.06
> >  DyCAST(No DAG) | 0.77±0.02 | 0.79±0.05| 0.95±0.03| 0.85±0.04| 0.98±0.04| 0.87±0.01
> >  DyCAST| 0.79±0.01| 0.90±0.13| 0.77±0.01| 0.91±0.11|0.75±0.04| 0.84±0.04
> >  DyCAST-CUTS+ | 0.81±0.02| 0.92±0.03 | 0.80±0.10| 0.89±0.12|0.80±0.01 | 0.85±0.04
> >
> > - **Ablation study on CausalTime dataset:**
> >    - We performed similar ablation experiments on the **CausalTime** dataset. This dataset provides a more realistic scenario and tests the generalizability of DyCAST's components in real-world-like dynamics. The results are summarized in **Table 9** (**Appendix D, page 23**).
> >
> > Methods|AQI| |Traffic| | Medical||
> > ---|---|---|---|---|---|---
> >  || AUROC | AUPRC | AUROC | AUPRC| AUROC | AUPRC
> > DYNOTEARS | 0.64±0.12| 0.41±0.08|0.81±0.08|0.33±0.12| 0.85±0.07| 0.32±0.13
> > w/o $\boldsymbol{S}_0$|0.73±0.02|0.61±0.03|0.65±0.01|0.57±0.03|0.74±0.01|0.63±0.10
> > w/o Latent State|0.80±0.07| 0.65±0.04|0.67±0.02| 0.60±0.05| 0.77±0.06| 0.70±0.02
> > DyCAST(No DAG) | 0.85±0.03| 0.70±0.03|0.68±0.05| 0.60±0.02 |0.74±0.01|0.69±0.03
> > DyCAST|0.85±0.02| 0.82±0.00|0.63±0.01| 0.70±0.01 |0.81±0.03| 0.74±0.03
> > DyCAST-CUTS+| 0.91±0.02 | 0.93±0.00 | 0.65±0.01 | 0.73±0.01|0.84±0.01| 0.77±0.03
> >
> > - The results confirm that $\boldsymbol{S}_0$ is **crucial** to DyCAST's success. Each row of $\boldsymbol{S}_0$ encodes complete information about how a variable is influenced by and influences others, making it the backbone of DyCAST's causal inference. The **hidden states** play a complementary role by reducing the dimensions of the intra- and inter-slice matrices. This compression not only makes the matrices more compact but also improves the overall performance of the model by mitigating noise.  We have included these experimental results in **Appendix D, page 22**, along with detailed explanations and discussions. These findings highlight the interplay between $\boldsymbol{S}_0$ and the hidden states in enhancing DyCAST's efficiency and accuracy.
> >
> > 3.*Minor Typos*
> >
> > **Response**: We thank the reviewer very much for pointing out the typos. For each point, we
> > - we revise the "NOTEATS" to "NOTEARS“  (Page 2, line 085)
> > - we revise the "DYNOTEATS" to "DYNOTEARS"  (Page 2, line 087);
> > - we revise the "$X^j_{t-k}$" to "$\boldsymbol{X}^j_{t-k}$" (Pages 6, line 124);
> > - we revise the "|" to "||" in Eq. (18) (Page 2, , line 283)
> > - we revise the "NTS-NOTEATS" to "NTS-NOTEARS"  (Page 8, line 398);
> > - we revise the "TEDCI" to "TECDI"  (Page 8, line 398);
> > - we revise the phrase "In future" to "In the future"  (Page 10, line 404);
> > - we revise the "Erdős–R'enyi" to "Erdős–Rényi" (Page 13, Line 657).
> >
> > 4.*A Few Other Minor Points*
> > **Response**:
> > - *The approach of ... be mentioned in related work.*
> >     - **Related work**:
> > we thank the reviewer for suggesting this nice work. We cite the recommended reference [1] in the main paper (lines 106-107, page 2).
> >
> > - *While I think the details in Appendix A ...reproducibility of the paper.*
> >     - **Implementation and Code**: to promote transparency and facilitate further research, we plan to release the code and implementation of DyCAST upon the acceptance of this paper. This will include detailed instructions and examples to help others replicate and build upon our work.
> > ---
> > **Reference**
> >
> > [1] Brouwer et al. (2021) Latent convergent cross mapping.

---

> > > ### Author Response · Authors · 2024-11-25
> > > **Repose, Part II/II**
> > >
> > > **Questions Part**
> > >
> > > 1.*My main question ... performed in the other experiments?*
> > >
> > > **Response**: We agree that ablation studies on real-world datasets are crucial to understanding the contributions of different components of DyCAST. To address this, we have conducted additional ablation experiments on both the **CausalTime** and **NetSim** datasets. The results, which are now included in **Tables 8 and 9 (Appendix D, page 23)**, demonstrate the impact of each component on the model’s performance. These experiments confirm that the **$\boldsymbol{S}_0$ matrix plays a critical role**, as it encodes the complete causal influence information for each variable, while the **hidden states effectively reduce the dimensionality of intra- and inter-slice matrices, enhancing compactness and improving performance**.
> > > We hope these additional experiments provide clarity on DyCAST's functionality. Further details can be found in the updated **Appendix D**.
> > >
> > > 2.*There is a brief ... baselines.*
> > >
> > > **Response**: we supplement numerical experiments to examine
> > > the running times of DyCAST and baseline methods across different numbers of variables **d={5,10,20,50,100}**. These additional experiments are presented in **Appendix C.7 (Figure 14, page 20)**. We show that DyCAST achieves competitive or superior efficiency compared to other methods.
> > >
> > > 3.*What are implementation ... p=1 case?*
> > >
> > > **Response**:Please see our response to point 1 for details on how to integrate DYCAST with CUTS+. To address the potential concern about computational overhead, we conducted a **running time analysis** for DyCAST, CUTS+, and their combined version (DyCAST-CUTS+).
> > >
> > > The results, summarized in the following,
> > > |Under d=20, T=8, N=500, epoch=1000|DyCAST|DyCAST(Not DAG)|DyCAST-CUTS+|
> > > ---|---|---|---|
> > > Running Times | 16.67min| 16.53min|17.34min
> > >
> > > we show that the integration with CUTS+ does not significantly increase the running time. DyCAST-CUTS+ remains computationally efficient, even with the added complexity of incorporating non-linear inter-slice relationships.
> > >
> > > - we would like to kindly highlight that, our fundamental model assumes fixed inter-slice relationships. For cases where $p > 1$, DyCAST flexibly handles higher-order autoregressive scenarios. We conducted experiments to evaluate DyCAST under unknown $p$. However, it should be noted that for DyCAST-CUTS+, when $ p > 1$, only a summary graph of the inter-slice dependencies can be provided, not a complete time graph.
> > >
> > > 4.*It seems like ... this doesn't work?*
> > >
> > > **Response**: In the current work, as defined in Eq. (3), we assume the inter-slice matrix $\boldsymbol{A}$ is static. This is a simple yet straightforward assumption that enables tractable modeling and inference.
> > >
> > > However, we acknowledge that in some real-world scenarios, the inter-slice matrix $\boldsymbol{A}$ may evolve dynamically over time. It's indeed straightforward to exploit Neural ODE for modeling the inter-slice dynamic, but learning two dynamics simultaneously is very difficult and can easily lead to data instability. Extending the framework to handle dynamic $\boldsymbol{A}$ is a valuable direction for future work.
> > >
> > > 5.*It seems surprising ... with faster dynamics?*
> > >
> > > **Response**: Based on the theoretical work by Fang et al. (2023) [1], causal graphs in real-world scenarios often exhibit central structures and thus inherently possess low-rank properties. This aligns with our findings that when  $r$ is set to an extremely low-rank value (**approximately 10% of the full rank**), the F1 score remains insensitive to $r$, as shown in **Figure 13, Appendix C.6, page 19**.
> > >
> > > However, as the rank of the intra-slice matrix increases, the F1 score starts to show a stronger dependency on the embedding dimension $r$. Specifically, increasing $r$ improves the F1 score in scenarios where the intra-slice matrix exhibits a higher rank.
> > >
> > > We supplement a detailed analysis in the **Figure 12 Appendix C.6 (page 19)**. We show that the F1 score is positively correlated with the size of the embedding dim $r$ under higher rank(**30% and 60%**) for the intra-slice structure. We believe this additional discussion provides clarity on the interplay between $r$ and model performance.

---

> > > > ### Author Response · Authors · 2024-11-25
> > > >
> > > > **Questions Part (continuing)**
> > > >
> > > > 6.*The methods for ... comparisons.*
> > > >
> > > > **Response**: While many baseline methods, aside from DYNOTEAR and its variants, do not produce a full-time graph, which is essential for evaluating dynamic causal models like DyCAST. For this reason, we specifically selected baselines like **DYNOTEAR, NTS-NOTEARS, and TECDI** which are capable of **producing full-time graphs**, to enable a fair evaluation of DyCAST on dynamic datasets.
> > > >
> > > > In the case of the NetSim and CausalTime datasets, we recognize the inconsistency in baseline performance, particularly when these datasets do not fully conform to a DAG structure. To address this, we have updated the comparison in the revised version of the paper. Specifically:
> > > >
> > > > - For NetSim, we have added results from **GC, CUTS+, LCCM, and DyCAST-CUTS+** to ensure a more comprehensive evaluation of DyCAST. Please see **Table 2, Section 4.2 (pages 9)** and **Table 6, Appendix C.10 (pages 21)** to verify it.
> > > >
> > > > - For CausalTime, we have also supplemented **TECDI**** as the baseline, providing a more well-rounded comparison. Please see **Table 3, Section 4.3 (pages 10)** to verify it.
> > > >
> > > > - Additionally, recognizing that the ground truth causal graphs for these two datasets may not strictly adhere to DAG structures, we conducted ablation studies where we removed the DAG constraints in DyCAST, resulting in a version we refer to as **DyCAST (Not DAG)**. This further explores the model’s flexibility in handling non-DAG structures.
> > > >
> > > > ---
> > > > **Reference**
> > > >
> > > > [1] Fang et al. (2023) On low-rank directed acyclic graphs and causal structure learning.

---

> ### Comment · Reviewer_AYt4 · 2024-11-26
>
> Thanks a lot for your detailed reply, which I think addressed most of my concerns. I have adjusted my rating accordingly.
>
> **On the Size of Tested Datasets**
> I agree with the authors that $d=300$ datasets are already fairly large. I do not think that the authors should make claims about "very large scale systems" or the like (and to my knowledge they don't), but find the inclusion of several real-world benchmarks to be more persuasive than larger synthetic datasets would be.
>
> **On the combination with other methods**
> Thanks for clarifying that CUTS+ is used everywhere. So, once again to clarify: DyCAST-CUTS+ can only recover the full graph in the $p=1$ case, right? This is okay, but is only really implicitly mentioned in Section 3 ("Notably, when the preceding variables span only a single time step, the summary graph is equivalent to the full-time DAG"); I think it would improve clarity to mention at the start (e.g., line 291) that "When $p=1$, we may take advantage of the summary graph coinciding with the inter-slice graph, using Granger-causal methods to learn inter-slice graphs."
>
> **Ablations**
> Thanks for including these. Regarding DyCAST w/o $S_0$: does this just mean that $z_0 \coloneqq \mathrm{Flatten}(\mathrm{ReLU}(W_0 P))$? I think this should be clarified.
>
> **On the Computational Cost of DyCAST+**
> Thanks for this table. Please consider adding this to an appendix.
>
> **Some Remaining Typos**
> - Line 275: "layres" should be "layers".
> - Throughout: Sometimes "DyCAST(Not DAG)" is written (no space between "DyCAST" and "(Not DAG)").
> - Line 461: "NTS-NOTEATS" should be "NTS-NOTEARS".
> - Line 461: Use `\citep` for (Gong et al., 2023)
> - Figure 9: The second plot is mislabeled as "F1 Score"; I think it's the SHD.
> - Figure 11: Once again I don't think either of these are showing F1 score, but the caption labels them as such.
> - Line 1082: "Obvious" should not be capital.
> - Appendix C.9: The title "Evaluation on Static Synthetic Data Across Algorithm with Separate Learning Tricky." doesn't make much sense, please consider revising it.

---

> ### Author Response · Authors · 2024-11-28
> **Response**
>
> We thank the reviewer so much for the reply and we really appreciate the reviewer for raising score.
>
> ---
>
> **Comment**
>
> 1.*I agree with the authors that ... than larger synthetic datasets would be.*
>
> **On the size of tested datasets**. We thank the reviewer for the suggestion. We conduct a new experiment on the DREAM-3 dataset, comprising five independent datasets, each with 10 time-series recordings from 100 genes across 21 time steps (see Table 11 in Appendix F). The results demonstrate that DyCAST remains highly competitive, significantly outperforming comparable methods such as DYNOTEARS and NTS-NOTEARS. Leveraging the nonlinear CUTS+ integration further boosts performance, we also find that integrating the nonlinear CUTS+ further improves performance leading to enhanced results.
>
> 2.*Thanks for clarifying that CUTS+ is used everywhere ... to learn inter-slice graphs.*
>
> **On the combination with other methods**. That's right that DyCAST-CUTS+ can only recover the full graph in the $p=1$ case. We now mention at the start (line 291) that "When $p = 1$, the summary graph coincides with the inter-slice graph, allowing the use of Granger-causal methods for learning inter-slice structure".
>
> 3.*Thanks for including these ... I think this should be clarified.*
>
> **Ablations**. We thank the reviewer for the suggestion. DyCAST without \( S_0 \) simply corresponds to $\boldsymbol{z}_0 := Flatten(ReLU( \boldsymbol{W}_0 \boldsymbol{P}))$. We have now included this clarification in **line 413** of Section 4.1 (Ablation Study) on page 8.
>
> 4.*Thanks for this table. Please consider adding this to an appendix.*
>
> **On the computational cost of DyCAST+** We thank the reviewer for the nice suggestion. We add an additional discussion on running time in **Table 6 (Appendix C.7, page 20)**.
>
> 5.*Some Remaining Typos*
>
> We thank the reviewer so much for pointing these typos out. For each of them,
>
> - We fix this as "layers" in line 275, page 6.
> - We add the missing space between "DyCAST" and "(Not DAG)".
> - We fix this as "NTS-NOTEARS" in line 464, page 9.
> - We add the "\citep" for (Gong et al., 2023) in line 464, page 9.
> - We fix the second and fourth plots for the mislabeled "F1 score" to "SHD" in Figure 9, Appendix C.4, page 17.
> - We fix this in Figure 11, Appendix C.5, page 18.
> - We fix this in lines 1082, page 21.
> - We revise the title of Appendix C.9 to "Challenges of evaluating separate learning on dynamic synthetic data".

---

### Official Review · Reviewer_4Eyh · 2024-11-04

**Soundness:** 2
**Presentation:** 3
**Contribution:** 2
**Rating:** 6
**Confidence:** 5

**Summary:**

The authors introduced the DyCAST algorithm to infer time-variant DAGs from nodal time series, considering both intra-slice and inter-slice dependencies among variables. They demonstrated its performance using both synthetic and real-world datasets and compared it with other causal discovery methods.

**Strengths:**

1. The authors addressed limitations of previous methods by considering both intra-slice and inter-slice dependencies among variables.
2. They offered clear and plausible theoretical motivation for their DyCAST algorithm.
3. The authors tested DyCAST's performance and compared it with previous methods.

**Weaknesses:**

1. The authors tested their DyCAST algorithm on datasets with a limited number of variables, which restricts its applicability in real-world scenarios.
2. While incorporating inter-slice dependencies is beneficial, the inference requires more data to be effective.
3. The algorithm relies on multiple independent realizations (runs) of system dynamics (i.e., nodal time series). For instance, in the first synthetic dataset, there are only d=5 variables and T=7 time steps (i.e. 7 snapshots of DAG), yet it requires N=500 independent realizations for inference. This is unrealistic, as many real-world situations may only provide one trial or a single trajectory of each variable.

**Questions:**

1. The authors used traffic as a motivation and test example; however, traffic networks are not DAGs. Similarly, brain networks also contain many bidirectional edges and feedback cycles. How would the algorithm address situations where the underlying network is a directed graph rather than a DAG?
2. How should the value of p (i.e., the order of time lag) be set in the algorithm? Is the ground truth for p assumed to be known for the inference process?
3. How much additional data is needed for inference when considering both intra- and inter-slice dependencies? If unlimited data were available, many other factors could be incorporated into causal discovery. Thus, the amount of required data is critical for addressing general inverse problems. The authors should specify how many realizations are necessary to achieve a given performance for a specific problem defined by the number of variables and time steps. Additionally, if p is unknown during the inference process, the algorithm may require even more data (i.e., independent realizations).

---

> ### Author Response · Authors · 2024-11-25
> **Response, Part I/II**
>
> We thank the reviewer so much for the valuable comments and appreciation of our work and efforts. Below we address your concerns and questions point by point.
>
> **Weaknesses Part**
>
> 1.*The authors tested their DyCAST algorithm ... in real-world scenarios.*
>
> **Response**: We acknowledge the importance of evaluating our method on large-scale variables of datasets to enhance the robustness of our results. Since real-world large-scale dynamic causal datasets for time series with ground truth are scarce, we conduct experiments on simulated data to ensure a fair comparison. Specifically, we evaluate datasets with $d = \{50, 100, 200, 300 \}$ variables, $T = 8$ time steps, and $ n = 500 $ samples in **Table 5 (Appendix C.1, page 15)**.
>  Methods |$d=50$ | | |$d=100$|  | |$d=200$ | | |$d=300$ | ||
> ---|---|---|---|---|---|---|---|---|---|---|---|---|
> | | TPR|SHD|F1| TPR|SHD|F1|TPR|SHD|F1|TPR|SHD|F1|
> DYNOTEARS| 1.00±0.00 | 2500±0.00 | 0.00±0.00|-|-|-|-|-|-|-|-|-|
> NTS-NOTEARS| 1.00±0.00 | 2500±0.00 | 0.00±0.00|-|-|-|-|-|-|-|-|-|
> TECDI| 1.00±0.00 | 2500±0.00 | 0.00±0.00|-|-|-|-|-|-|-|-|-|
> DyCAST| 0.89±0.02 | 9.01±3.14 | 0.73±0.05 |0.87±0.12|16.00±4.17|0.68±0.08|0.61±0.10 |18.89±5.53|0.67±0.04|0.43±0.04|49.11±7.14|0.56±0.01|
>
> DYNOTEARS, NTS-NOTEARS, and TECDI fail to converge with 50 variables, misidentifying all causal edges, while DyCAST achieves an average F1 score of 0.73. At 300 variables, the compared baselines remain non-convergent, but DyCAST maintains an F1 score of 0.56 and a low SHD (49.11±7.14), demonstrating scalability and robustness in large-scale causal structures.
>
> 2.*While incorporating inter-slice ... to be effective.*
>
> **Response**:
> We thank the reviewer so much for this insightful question. To address this concern, we also conduct an experiment on datasets with $d = 20$ variables, $T = 8$ time steps, and over **different sample sizes** ($N = {400, 300, 200, 100, 50, 20}$). DyCAST uses observation data from the current moment ($t$) and the past $p$ ($\boldsymbol{X}{t-p:t-1}$) time points, consistent with other baselines. The corresponding discussion is presented in **Figure 7 (Appendix C.2, page 16)**. Quantitative experiments show that DyCAST achieves an **F1 score of ~0.4** and **SHD <30** with as few as **20 samples**, whereas baselines fail to identify any valid causal relationships, demonstrating DyCAST's ability to handle limited data effectively.
>
> 3.*The algorithm relies on ... single trajectory of each variable.*
>
>
>
> **Response**: To address your concern, we experiment to examine the scalability and efficiency. We run DyCAST across datasets over **different sample sizes** (**N = {400, 300, 200, 100, 50, 20}**) and **length** (**T=8, 16, 32, 64, 128**). These additional experiments are presented in **Appendix C.2** and **C.3** (pages 16 to 17). Specifically,
> - we show that DyCAST achieve an **average F1 score** of approximately **0.4** even with as few as **20 samples** in **Figure 7**. In contrast, baseline models were unable to identify any valid causal relationships under similar low-sample conditions. The potential of DyCAST to effectively handle limited trajectories has been thoroughly demonstrated.
>
> - We show that,  despite the growing complexity of internal dynamics with longer time sequences, DyCAST consistently achieved an **F1 score** above **0.6** across all tested time step lengths, **with a fixed sample size of 500** in **Figure 8**.The potential of DyCAST to effectively handle long temporal dynamics has been thoroughly demonstrated.
>
> - Through extensive experiments, we determined that the model achieves **80%** accuracy when the ratio of parameters to data is approximately **0.025%**. In practice, we observed that for sufficiently simple causal relationships, the model performs well even with a lower ratio.

---

> ### Author Response · Authors · 2024-11-25
> **Response, Part II/II**
>
> **Questions Part**
>
> 1.*The authors used traffic ... than a DAG?*
>
> **Response**: We agree that traffic networks and brain networks might not be DAGs in some cases. Fortunately, our method can be easily extended to handle directed graphs (non-DAGs). By simply removing the manifold constraint terms in the **Eq.(9) (line 201, page 4)**, DyCAST can infer the dynamics of directed graphs effectively.
>
> We conducted experiments using DyCAST (Not DAG) on the NetSim and CausalTime datasets. These additional experiments are presented in **Table 2 (Section 4.1, page 9), Table 3 (Section 4.2, page 10), and Table 6 (Appendix C.10, page 21)**.We show that DyCAST (Not DAG) achieves impressive AUROC values for both datasets. However, removing the DAG constraint introduces spurious bidirectional edges in the inferred causal graph, slightly reducing the AUPRC metric.
>
> 2.*How should the value ... inference process?*
>
> **Response**: We sincerely appreciate the reviewer's insightful comment regarding the selection of the time-lag order $p$. In fact, DyCAST does not require prior knowledge of the true lag order in the data. To demonstrate this, we evaluated DyCAST with $p=3$ on simulated data where the true lag order was $p_\text{true}=1$ (**Figure 10, Appendix C.5, page 18**). The results showed that DyCAST identified only three very small spurious entries (less than 0.1) in the estimated $A_2$, and no spurious entries in $A_3$. This suggests that DyCAST is robust to the choice of $p$, and selecting a slightly larger lag order does not significantly affect the final inference results.
>
> Further details of this experiment can be found in the **Appendix C.5**. Additionally, we provide two practical strategies for determining $p$ in the Appendix:  (1). Using the Bayesian Information Criterion (BIC) based on the model loss to select \$p$ as shown in **left panel** of Figure 11, **Appendix C.5, page 18**. (2). Estimating $p$ by analyzing the maximum entry of the inferred $A_p$ matrices ($\max(A_p) $) as shown in **right panel** of **Figure 11, Appendix C.5, page 18**). These approaches can guide users to select an appropriate lag order without prior knowledge of $p_\text{true}$.

---

### Author Response · Authors · 2024-11-25
**General Response to All Reviewers and Summary of Changes of the Revision**

We thank all reviewers a lot for the valuable comments and insightful questions and suggestions.

To answer questions and address the concerns of the reviewers, we make the corresponding changes in the rebuttal revision, which will be summarized as follows: additional experiments for scalability, efficiency, and robustness; extension of DyCAST. For clarity, all changes in the revised manuscript are highlighted in blue.

***
### Additional Experiments for Scalability, Efficiency, and Robustness

As suggested by Reviewer **4Eyh**, **AYt4**, **gcbX** and **e4ie**, we supplement experiments to examine the scalability, efficiency, and robustness of the DyCAST. These additional experiments are presented in **Appendix C** (pages 15 to 22) and **Appendix E**(page 23). Specifically:

1. Scalability analysis over datasets with large-scale variables (in **Table 5, Appendix C.1, page 15**). We show that DyCAST still displays an average **F1 score> 0.5** when the number of variables **d=300**, which is much higher than the number of variables that other baseline methods can handle, as shown in **Table 5**. Correspondingly, DYNOTEARS and its variants fail to identify causal relationships when the variable count reaches $d = 50$ and does not converge for $d > 50$. Thus, the potential of DyCAST to effectively handle large-scale dynamic systems has been thoroughly demonstrated.

2. Sample efficiency analysis over datasets with different sample sizes (in **Figure 7, Appendix C.2, page 16**). We show that DyCAST still displays an average **F1 score> 0.4** and an average **SHD < 30** when the number of sample sizes **N=20**, as shown in **Figure 7**. In contrast, with a sample size below 300, DYNOTEARS and its variants exhibit a sharp decline in causal recognition, with an average F1 score around 0.1 and SHD exceeding 200, rendering effective recognition infeasible. Thus, the potential of DyCAST to effectively handle limited trajectories has been thoroughly demonstrated.

3. Temporal generalization analysis over datasets with different lengths (in **Figure 8, Appendix C.3, page 17**). We show that DyCAST displays an average **F1 score> 0.5** and an average **SHD < 20** when the length of time steps **T=128**, as shown in **Figure 8**. In contrast, DYNOTEARS and its variants fail to converge when time steps exceed 64, with SHD surging to 100 (out of 100 total edges in this experiment). Thus, the potential of DyCAST to effectively handle long temporal dynamics has been thoroughly demonstrated.


5. Robustness to noise analysis over datasets with different strengths and types of noise (in **Figure 9, Appendix C.4, page 17**). We show that DyCAST exhibits robustness to noise intensity and type, as their effects are primarily limited to the initial intra-slice and inter-slice learning phases, with **minimal impact** on subsequent intra-slice accuracy due to the Neural ODE's ability to capture underlying dynamics, as shown in **Figure 9**. This demonstrates the model's resilience in learning robust temporal dependencies under noisy conditions.

6. Adaptability analysis over datasets with unknown autoregressive order (in **Figure 10, Figure 11, Appendix C.5, page 18**). We show that DyCAST can effectively estimate inter-slice entries even without prior knowledge of the autoregressive order $p$, highlighting its adaptability to unknown temporal dependencies.

7. More ablation studies over synthetic and real-world datasets (in **Table 1 (Section 4.1, page 8), Table 8, Table 9, (Appendix D, pages 22 to 23)**). We conducted extensive ablation studies on real and synthetic datasets, analyzing the impact of $\boldsymbol{S}_0$ and latent states to uncover the underlying factors driving DyCAST's performance.  Each row of $\boldsymbol{S}_0$ encodes how each intra-slice variable is influenced by others and, in turn, influences them. Without the concatenation operation in $\boldsymbol{S}_0$, DyCAST remains incomplete after the encoder, making this step crucial for its performance. Additionally, leveraging $\boldsymbol{S}_0$ to derive a more compact hidden state representation can further enhance DyCAST's computational efficiency and effectiveness.

---

> ### Author Response · Authors · 2024-11-25
> **General Response to All Reviewers and Summary of Changes of the Revision  (continuing)**
>
> ***
> ### Extension of DyCAST
> ***
>
> As suggested by Reviewer **AYt4**, we discuss the extension of DyCAST in Section 3.3 (starting from line 289, page 6 of the revision). Specifically, we show that:
>
> 1. Our DyCAST modeling framework is flexible, which can be easily incorporated into other causal discovery approaches, particularly those for time series, with slightly modifying our **Eq.(3)** into:
>
> $$\boldsymbol{X}^i_t = \boldsymbol{X}_t \boldsymbol{W}_t + \varphi(\boldsymbol{X}{t-p:t-1}) + \boldsymbol{Z}^i_t$$
>
> where $\varphi$ can be a complex nonlinear function.
>
> Following this extension, we show that DyCAST can be introduced to CUTS+, which is a nonlinear model specifically designed to construct $\varphi$. To examine the effects of DyCAST-CUTS+ on complex real-world datasets in the causal discovery task, we also run DyCAST-CUTS+ on NetSim and CausalTime datasets. DyCAST-CUTS+ achieves the highest average performance on both NetSim and CausalTime, consistently providing performance improvement over the original CUTS+. We report the results in **Table 2 (Section 4.2, page 9), Table 3 (Section 4.3, page 10), and Table 6 (Appendix C.10, page 21)**.

---

### Meta-Review · Area_Chair_e26L · 2024-12-19

**Metareview:**

This paper performs causal discovery for time-varying dynamic causal structures. In particular, it employs a constrained latent neural ODE to model the dynamics and its evolution. Most of the reviewers evaluated the paper positively. There was a question about the clarity of the presentation, which I hope can be addressed in the camera ready version of the paper. I would also hope that the authors can discuss whether and when the causal structure model considered in this paper is identifiable.

**Additional Comments On Reviewer Discussion:**

Most of the comments and concerns from the reviewers are addressed. There was one remaining question about the clarity of the presentation, which I hope can be addressed in the camera ready version of the paper.

---

### Decision · Program_Chairs · 2025-01-22

Accept (Poster)